# GPT-FL: Generative Pre-trained Model-Assisted Federated Learning

## Abstract

In this work, we propose `GPT-FL`, a generative pre-trained model-assisted federated learning (FL) framework. At its core, `GPT-FL` leverages generative pre-trained models to generate diversified synthetic data. These generated data are used to train a downstream model on the server, which is then fine-tuned with private client data under the standard FL framework. We show that `GPT-FL` consistently outperforms state-of-the-art FL methods in terms of model test accuracy, communication efficiency, and client sampling efficiency. Through comprehensive ablation analysis, we discover that the downstream model generated by synthetic data plays a crucial role in controlling the direction of gradient diversity during FL training, which enhances convergence speed and contributes to the notable accuracy boost observed with `GPT-FL`. Also, regardless of whether the target data falls within or outside the domain of the pre-trained generative model, `GPT-FL` consistently achieves significant performance gains, surpassing the results obtained by models trained solely with FL or synthetic data.

## 1 Introduction

Federated learning (FL) is a privacy-preserving machine learning paradigm that allows a collection of clients to collaboratively train a machine learning model without sharing their private data Zhang et al. (2021). Most existing FL studies such as McMahan et al. (2016); Bonawitz et al. (2019) follow the standard FL architecture, where each participating client trains a local model using its own private data and a central server aggregates these locally trained models to update a global model and send it back to the clients for the next round of training. However, although many efforts have been made Sahu et al. (2018); Karimireddy et al. (2019); Reddi et al. (2020), the performance of standard FL is still constrained by client drift caused by the heterogeneity in private data distribution across the clients.

To enhance the performance of FL, recent studies propose to incorporate data collected from public spaces such as the internet into the FL process Lin et al. (2020); Li et al. (2021); Itahara et al. (2020); Cho et al. (2022). However, the performance of such public data-based approaches is heavily dependent on the quality of the collected public data. Unfortunately, obtaining the desired public data can be extremely challenging in practice and there is a lack of principled guidance on how to obtain them. To address the issues of public data-based approaches, FL methods based on synthetic data emerge Zhang et al. (2022); Zhu et al. (2021); Pi et al. (2022); Wijesinghe et al. (2023). In Zhang et al. (2022); Zhu et al. (2021), a generative model is trained through knowledge distillation (KD) and the synthetic data are generated from the generative model in an *interleaved* manner *throughout* the federated training iterations. Unfortunately, these approaches are confronted with two limitations: (1) since the training of the generative model and the federated training process interleave, the quality of the synthetic data generated by the generative model before it converges can be extremely unstable. Such low-quality synthetic data would in turn jeopardize the federated training process; (2) given that KD requires clients to report model weights as teachers to transfer knowledge, they are incompatible with secure aggregation protocols Bonawitz et al. (2017); So et al. (2021), which limits their privacy guarantee compared to standard FL.

In this work, we propose `GPT-FL`, a generative pre-trained model-assisted FL framework that effectively addresses the issues of existing methods. The key idea behind `GPT-FL` is to leverage the knowledge from the generative pre-trained models and to *decouple* synthetic data generation from

Table 1: Comparison of `GPT-FL` with existing FL methods.

| | External Data | Limited to Smaller Client Model | Generate Data during FL | Data Generator Location | Client Access to Public/Generated Data | Support Data Modality | Compatibility with Secure Aggregation |
|---|---|---|---|---|---|---|---|
| FedAvg McMahan et al. (2016)
FedOpt Reddi et al. (2020)
FedProx Sahu et al. (2018)
SCAFFOLD Karimireddy et al. (2019) | No | No | N/A | N/A | N/A | Any | Yes |
| FedDF Lin et al. (2020)
DS-FL Itahara et al. (2020)
Fed-ET Cho et al. (2022) | Public Data | No | N/A | N/A | Not Required
Required
Not Required | Any
Any
Any | No |
| MOON Li et al. (2021) | No | No | N/A | N/A | Not Required | Only Image | Yes |
| FedGen Zhu et al. (2021)
FedFTG Zhang et al. (2022)
DynaFed Pi et al. (2022)
**GPT-FL (Ours)** | Generated Data


 | Yes


**No** | Yes


**No** | Client
Server
Server
**Server** | Required
Not Required
Not Required
**Not Required** | Only Image
Only Image
Only Image
**Any** | No
No
Yes
**Yes** |

the federated training process. Specifically, `GPT-FL` prompts the generative pre-trained models to generate diversified synthetic data. These generated data are used to train a downstream model on the server in the centralized manner, which is then fine-tuned with the private client data under the standard FL framework. By doing this, the proposed `GPT-FL` is able to combine the advantages of previous methods while addressing their limitations.

The proposed `GPT-FL` exhibits multifold merits compared to prior arts (Table 1): (1) In contrast to public data-based FL methods, `GPT-FL` gets rid of the dependency on the availability of the desired public data, offering much more flexibility in its applications. (2) Compared to other generative data-based approaches, the leverage of generative pre-trained models and the decoupling between synthetic data generation from the federated training process make the generated synthetic data in `GPT-FL` not impacted by private data distribution on the clients and the structure of the model to be trained. (3) By leveraging the computational resources on the server, `GPT-FL` provides a much more efficient way to utilize external data by incorporating them into the pre-training of the downstream model, which significantly reduces the communication and computation costs of FL. (4) The generation of downstream models using synthetic data takes place on the server. As such, it thereby eliminates the need for clients to bear any additional computational burden. (5) Lastly, as `GPT-FL` does not alter the standard FL framework, it is fully compatible with secure aggregation protocols as in standard FL methods. More importantly, `GPT-FL` does not introduce any additional hyper-parameters beyond the standard FL framework. This significantly simplifies the hyper-parameter optimization process, making `GPT-FL` much more practically useful.

We evaluate the performance of `GPT-FL` by comparing it against state-of-the-art FL methods under three categories: standard FL methods, public data-based methods, and generated data-based methods on five datasets that cover both image and audio data modalities. We highlight five of our findings: (1) `GPT-FL` consistently outperforms state-of-the-art FL methods under both low and high data heterogeneity scenarios with significant advantages in communication and client sampling efficiency. (2) Under a zero-shot setting, *i.e.* no real-world data is available, the downstream model after centralized training with synthetic images as part of `GPT-FL` achieves higher performance compared to the global model based on standard FL training with private data. On the contrary, the centralized training with synthetic audio performs worse than FL setups due to the impact of data modality and the quality of the generative pre-trained models. (3) GPT-FL does not fully rely on generated data. Regardless of whether the target data falls within or outside the domain of the pre-trained generative model, `GPT-FL` can largely improve model performance beyond relying solely on private data in a standard FL framework. (4) The downstream model generated by synthetic data controls gradient diversity during FL training, improving convergence speed and leading to significant accuracy gains with `GPT-FL`. (5) `GPT-FL` effectively leverages existing pre-trained downstream models to improve performance in the FL setting, similar to methods under the standard FL framework.

## 2 RELATED WORK

**Standard Federated Learning.** In standard federated learning (FL), clients perform local model training on their private data whereas the central server aggregates these locally trained models to update a global model, which is then sent back for the next round of training. To enhance privacy, Secure Aggregation (SA) protocols Bonawitz et al. (2017); So et al. (2021) have been proposed to encrypt each model update and reveal only the sum of the updates to the server. However, the

performance of FL is jeopardized by client drift which is caused by the heterogeneity of private data distribution. To tackle this issue, FedProx Sahu et al. (2018) introduces a proximal term to the local subproblem to constrain the local update closer to the global model; SCAFFOLD Karimireddy et al. (2019) leverages a variance reduction technique to mitigate the effect of drifted local updates; and FedOpt Reddi et al. (2020) proposes to update the global model by applying a gradient-based server optimizer to the average of the clients' model updates.

**FL with Public Data.** To further mitigate client drift, recent studies propose to utilize public data (e.g., collected from the internet) in the process of federated training. For example, FedDF Lin et al. (2020) leverages public data at the server to aggregate client models through knowledge distillation (KD). DS-FL Itahara et al. (2020) proposes a similar approach based on semi-supervised FL. MOON Li et al. (2021) proposes to use contrastive loss to further improve the performance. Fed-ET Cho et al. (2022) introduces a weight consensus distillation scheme using public data to train a large server model with smaller client models. However, utilizing public data for FL has several limitations: the performance of FL heavily relies on the selected public data. However, it is unclear to which extent should the publish data be related to the training data to guarantee effective knowledge distillation, making it challenging to find appropriate public data for every use case Stanton et al. (2021); Alam et al. (2022); Zhang et al. (2022). Moreover, the involvement of KD requires clients to send model weights to the server. This requirement makes it incompatible with secure aggregation protocols, making them vulnerable to backdoor attacks Wang et al. (2020). Furthermore, some proposed methods Li et al. (2021); Lin et al. (2020) require clients to process the public data. Such requirement adds an extra computational burden to clients.

**FL with Synthetic Data.** To address the issues of public data-based approaches, FL methods based on synthetic data have been proposed Zhang et al. (2022); Zhu et al. (2021); Pi et al. (2022); Wijesinghe et al. (2023). In particular, FedGen Zhu et al. (2021) proposes to train a lightweight generator on the server using an ensemble of local models in a data-free manner. The generator is then sent to the clients to regularize local training. FedFTG Zhang et al. (2022) trains a GAN-based generator where the global model acts as the discriminator. The generated data are then used to fine-tune the global model on the server after model aggregation. However, training of the generator relies heavily on the global model, which can lead to poor performance under high data heterogeneity. Additionally, the quality of training the generator is impacted by the structure of the global model Kim et al. (2022), making the quality of the synthetic data unstable during training. Furthermore, these approaches are limited to image-related tasks, restricting their applicability to other data modalities. Specifically, both FedGen and FedFTG rely on training MLP-based or GAN-based lightweight generator networks to ensemble user information in a data-free manner, where the lightweight generator may have limitations in generating high-fidelity data. In addition, the MLP-based model is impractical to model temporal structures to signals such as audio and speech. Finally, some approaches Zhang et al. (2022); Zhu et al. (2021) could not support secure aggregation protocols due to the KD-based training, which could compromise the privacy of client data. As an alternative, DynaFed Pi et al. (2022) proposes to generate synthetic data via gradient inversion by applying multi-step parameter matching on global model trajectories and using the synthesized data to help aggregate the deflected clients into the global model. However, using gradient inversion for generating synthetic data could encounter limitations when dealing with high-resolution images Huang et al. (2021). In addition, this approach could not be directly used for other data modalities such as audio Dang et al. (2021). In this work, we propose GPT-FL as a solution to address these limitations.

## 3 GPT-FL: GENERATIVE PRE-TRAINED MODEL-ASSISTED FEDERATED LEARNING

The overall architecture of GPT-FL is illustrated in Figure 1. As shown, GPT-FL consists of four steps. First, prompts are created based on the label names at the server. These prompts are then utilized to guide the generative pre-trained models to generate synthetic data. The server uses these generated synthetic data to train a downstream model and distributes the trained model to the clients. Lastly, the clients use the trained model as the starting point, and finetune the model with their private data under the standard FL framework until it converges. In the following, we describe the details in each step.

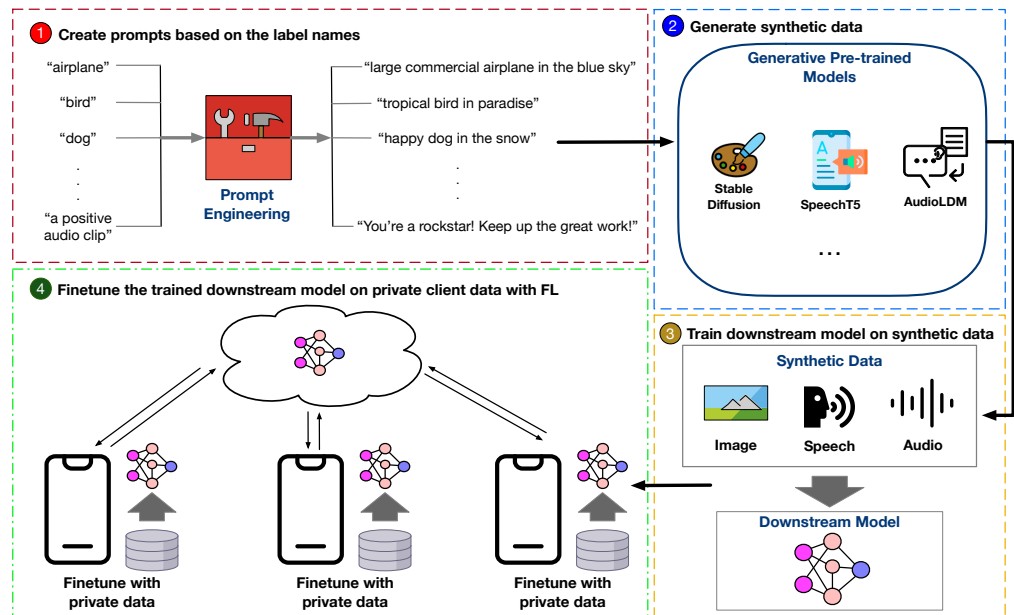

Figure 1: Overview of GPT-FL.

## 3.1 CREATE PROMPTS BASED ON LABEL NAMES

As the first step of GPT-FL, a prompt that describes the desired content of the data is required to guide the synthetic data generation process. To do so, GPT-FL requires the clients to provide the set of label names[1] of their private local data to generate prompts. However, prior research Shipard et al. (2023); He et al. (2022) shows that using only label names to generate prompts could restrict the quality and diversity of the generated synthetic data. Moreover, in FL, the server does not have access to detailed descriptions of the private data. To address these issues, GPT-FL incorporates large language models (LLMs) such as GPT-3 to expand each input class's details and use them as prompts for synthetic data generation. As an example, for the label name "airplane", GPT-FL uses the following query for the LLM to generate the prompt as follows:

```
Q: " _ _ _ _ airplane _ _ _ _" Please fill in the blank and
   make it as a prompt to generate the image
A: Large commercial airplane in the blue sky.
```

Moreover, inspired by Shipard et al. (2023), we randomly set the unconditional guidance scale of the Stable Diffusion model between 1 and 5 to further enrich the data diversity. In addition to the aforementioned techniques, it is worth noting that GPT-FL is flexible and compatible with other prompt engineering techniques that can be used to generate diversified synthetic data.

It should be noted that GPT-FL can employ Invertible Bloom Lookup Tables (IBLT) to encode label names before sending them to the server so that the label information of each client is not leaked to the server Gascón et al. (2023). Specifically, each client locally encodes its unique label names into IBLT, a probabilistic data structure that can encode items in an open domain efficiently. The server linearly aggregates these IBLTs via the secure aggregation Bonawitz et al. (2016) and decodes the aggregated table for the union of unique label names without revealing individual label information. More details about IBLT in GPT-FL are provided in Appendix A via an illustrative experiment.

## 3.2 GENERATE SYNTHETIC DATA FROM GENERATIVE PRE-TRAINED MODEL

Next, the generated prompts are used as the inputs to the generative pre-trained models to generate synthetic data. In this work, we utilize the state-of-the-art Latent Diffusion Model Rombach et al. (2021) loaded with Stable Diffusion V2.1 weights to generate synthetic images for image-based FL

---

[1]To protect user data privacy in FL setting, GPT-FL only requests the set of distinct label names instead of detailed label name distributions, and generates a uniform number of prompts for each label name.

applications; and we utilize the state-of-the-art SpeechT5 model Ao et al. (2021) for text-to-speech and AudioLDM model Liu et al. (2023) for text-to-audio to generate synthetic speech and audio data, respectively. It should be noted that the proposed GPT-FL is a general framework that supports other generative pre-trained models and data modalities beyond images and audio.

### 3.3 TRAIN DOWNSTREAM MODEL ON GENERATED SYNTHETIC DATA

With the generated synthetic data, GPT-FL trains a downstream model on the server in a centralized manner, and distributes the trained model to the clients participated in FL. This trained model acts as the initialized model for the following federated training process. One note should be emphasized from our empirical experiences is that training with synthetic data is prone to overfitting, as synthetic data tend to be highly patternized compared to real data. To mitigate the effects of overfitting, we adopt relatively large weight decay hyperparameters and small learning rates compared to training with real data. The detailed hyper-parameter selections are listed in Appendix A.

### 3.4 FINETUNE TRAINED DOWNSTREAM MODEL ON PRIVATE CLIENT DATA WITH FL

Lastly, the clients use the trained model distributed from the server as the starting point, and finetune the model with their private data under the standard FL framework until the finetuning converges. As such, GPT-FL does not alter the standard FL framework, making it fully compatible with secure aggregation protocols as in standard FL methods. More importantly, unlike existing generated data-based approaches Zhang et al. (2022); Zhu et al. (2021); Pi et al. (2022), GPT-FL does not introduce any additional hyper-parameters beyond the standard FL framework. This significantly simplifies the hyper-parameter optimization process, making GPT-FL much more practically useful.

## 4 EXPERIMENTS

**Datasets, Models, and Tasks.** We evaluate the performance of GPT-FL on five datasets from three FL applications: image classification, speech keyword spotting, and environmental sound classification. For image classification, we conduct experiments on CIFAR-10, CIFAR-100 Krizhevsky (2009), and Oxford 102 Flower Nilsback & Zisserman (2008) using ConvNet Pi et al. (2022), ResNet18, ResNet50 He et al. (2015), and VGG19 Simonyan & Zisserman (2014). Among them, CIFAR-10 and CIFAR-100 contain images from diverse objects whereas Oxford 102 Flower only contains images of flowers but with higher resolutions for fine-grained classification. For audio-related tasks, we choose the Google Command speech dataset Warden (2018) for keyword spotting and ESC-50 dataset Piczak for environmental sound classification. We followed the previous work Zhang et al. (2023) to use the same model for these two datasets. More detailed information about the data-preprocessing method and model setups is described in Appendix.

**Data Heterogeneity.** For CIFAR-10 and CIFAR-100, the training dataset is partitioned heterogeneously amongst 100 clients using the Dirichlet distribution $Dir_K(\alpha)$ with $\alpha$ equal to 0.1 and 0.5 following the previous work Cho et al. (2022). With the same method, we partition Flowers102 into 50 subsets due to its relatively small size. For audio datasets, Google Speech Command is partitioned over speaker IDs, making the dataset naturally non-IID distributed. It contains a total of 105,829 audio recordings collected from 2,618 speakers. The training set includes the recordings from 2,112 speakers and the test set includes the rest. To create non-IID data distributions on ESC-50, we followed the previous work Zhang et al. (2023) to partition ESC-50 into 100 subsets using $Dir_K(\alpha)$ with $\alpha$ equal to 0.1.

**Baselines and Evaluation Metrics.** We compare GPT-FL against three categories of baselines: 1) standard FL methods without the use of public or generated synthetic data – FedAvg, FedProx, and Scaffold; 2) FL methods that involve the use of public data – MOON, FedDF, DS-FL, and Fed-ET; and 3) FL methods that utilize generated synthetic data – FedGen and DynaFed[2]. We use the test accuracy of the trained model as our evaluation metric. We run experiments with three different random seeds and report the average and standard deviation. The details of the hyper-parameter selection of each dataset and experiment are described in Appendix.

---

[2]We did not compare with FedFTG because its code is not open-source, and we could not reproduce their results following the paper.

[3]Zhu et al. (2021); Pi et al. (2022) only reported results on ConvNet. We tested these two methods on VGG19 but they are not converged.

Table 2: Model accuracy comparison between `GPT-FL` and existing FL methods. For public data-based methods MOON, FedDF, DS-FL and Fed-ET, the results on CIFAR-10 and CIFAR-100 are obtained from Cho et al. (2022), and the results on Flowers102 are marked as N/A given the practical challenge on finding a set of suitable public data that can boost its performance.

| Method | Training Model | High Data Heterogeneity ($\alpha = 0.1$) | | | Low Data Heterogeneity ($\alpha = 0.5$) | | |
|---|---|---|---|---|---|---|---|
| | | CIFAR-10 | CIFAR-100 | Flowers102 | CIFAR-10 | CIFAR-100 | Flowers102 |
| FedAvg | | 71.19 (± 0.27) | 30.21 (± 0.32) | 30.30 (± 0.16) | 74.82 (± 0.23) | 33.12 (± 0.13) | 34.75 (± 0.90) |
| FedProx | VGG19 | 72.45 (± 0.13) | 31.51 (± 0.11) | 33.23 (± 0.24) | 75.24 (± 0.19) | 33.64 (± 0.08) | 40.56 (± 0.19) |
| SCAFFOLD | | 75.12 (± 0.20) | 30.61 (± 0.57) | 26.75 (± 0.50) | 78.69 (± 0.15) | 34.91 (± 0.61) | 33.21 (± 0.41) |
| MOON | | 75.68 (± 0.51) | 33.72 (± 0.89) | N/A | 81.17 (± 0.41) | 42.15 (± 0.72) | N/A |
| FedDF | VGG19 | 73.81 (± 0.42) | 31.87 (± 0.46) | N/A | 76.55 (± 0.32) | 37.87 (± 0.31) | N/A |
| DS-FL | | 65.27 (± 0.53) | 29.12 (± 0.51) | N/A | 68.44 (± 0.47) | 33.56 (± 0.55) | N/A |
| Fed-ET | | 78.66 (± 0.31) | 35.78 (± 0.45) | N/A | 81.13 (± 0.28) | 41.58 (± 0.36) | N/A |
| FedGen | ConvNet[3] | 42.05 (± 0.93) | 26.64 (± 0.66) | Not Converged | 54.86 (± 0.13) | 34.03 (± 0.42) | Not Converged |
| DynaFed | | 71.59 (± 0.10) | 36.08 (± 0.15) | Not Converged | 75.66 (± 0.21) | 43.82 (± 0.30) | Not Converged |
| **GPT-FL** | VGG19 | **82.16 (± 0.13)** | **47.80 (± 0.32)** | **70.56 (± 0.34)** | **82.17 (± 0.20)** | **48.39 (± 0.17)** | **74.84 (± 0.43)** |
| | ConvNet | **72.62 (± 0.24)** | **42.66 (± 0.19)** | **37.91 (± 0.43)** | **77.18 (± 0.21)** | **47.89 (± 0.28)** | **48.61 (± 0.51)** |

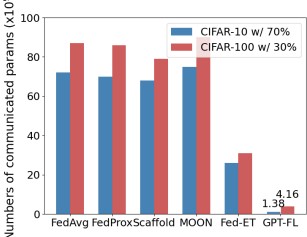

Figure 2: Communication costs of standard FL methods, public data-based methods and `GPT-FL` to achieve the target test accuracy.

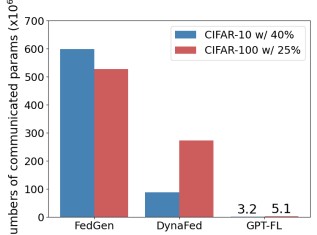

Figure 3: Communication costs of generated data-based methods and `GPT-FL` to achieve the target test accuracy.

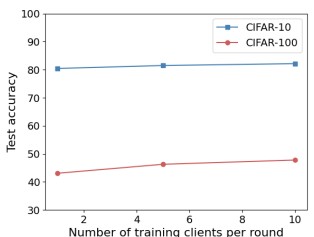

Figure 4: Test accuracy of `GPT-FL` for CIFAR-10/100 under different client sampling rates.

## 4.1 Performance Comparison with State-of-the-Art FL Methods

First, we compare the performance of `GPT-FL` with state-of-the-art FL methods. To enforce fair comparisons, in this experiment, we choose to evaluate on the three image datasets (CIFAR-10, CIFAR-100 and Flowers102) since baseline methods MOON, FedGen and DynaFed only support image data. Moreover, we used the same models (VGG19 and ConvNet) and experiment settings as previous work Cho et al. (2022); Pi et al. (2022). In each communication round, We randomly sample 10 clients from 100 clients for CIFAR and use all 50 clients for Flowers102. We choose FedAvg as the FL optimizer. All the training starts from random initialization and total number of communication rounds is set to 500.

**Overall Performance.** Table 2 summarizes our results. We make three key observations: (1) `GPT-FL` consistently outperforms all the baselines we selected in Table 2 under both low and high data heterogeneity scenarios across all three datasets. (2) In direct comparison with state-of-the-art generated data-based FL methods, although FedGen and DynaFed perform reasonably well on CIFAR-10 and CIFAR-100, they do not converge on Flowers102 whose images have higher resolutions than CIFAR. Moreover, both FedGen and DynaFed fail to converge when training a larger VGG19 model on Flowers102 and even lower-resolution CIFAR-10/100. In contrast, `GPT-FL` not only converges but also achieves state-of-the-art accuracy on Flowers102. More importantly, `GPT-FL` is able to support larger model, and its accuracy is significantly higher than the smaller ConvNet. (3) For Flowers102, as both public data-based and generated data-based FL methods are confronted with challenges, the only viable options are standard FL methods and `GPT-FL`. As shown, with the same model, `GPT-FL` outperforms standard FL methods by a significant margin.

**Communication Efficiency.** Besides model accuracy, we also compare the communication costs of `GPT-FL` with existing FL methods on CIFAR-10/100 under high data heterogeneity, where communication cost is measured as the total number of model parameters communicated between the server and clients during federated training until reaching a target model test accuracy. Specifically, Figure 2 shows the communication cost comparison between standard FL methods, public data-based

Table 3: Accuracy performance of the generated downstream model and standard FL on benchmark datasets. "1x Synthetic" represents the size of synthetic data is one time as the real data.

| | Dataset | 1x Synthetic | 2x Synthetic | 3x Synthetic | FedAvg | FedOpt |
|---|---|---|---|---|---|---|
| Image Data | CIFAR-10 | 61.48 (± 0.08) | 67.41 (± 0.40) | **75.65 (± 0.09)** | 64.48 (± 0.13) | 72.68 (± 0.22) |
| | CIFAR-100 | 24.70 (± 0.00) | 33.41 (± 0.01) | **41.76 (± 0.03)** | 25.89 (± 0.67) | 20.85 (± 0.14) |
| | Flowers102 | 24.94 (± 0.57) | 28.26 (± 0.14) | **31.29 (± 0.18)** | 30.30 (± 0.16) | 26.43 (± 0.09) |
| Audio Data | Google Command | 24.78 (± 0.04) | 25.65 (± 0.07) | 26.24 (± 0.01) | 73.68 (± 0.49) | **83.01 (± 0.23)** |
| | ESC-50 | 6.89 (± 0.29) | 8.68 (± 0.35) | 12.72 (± 0.31) | 22.76 (± 1.01) | **32.49 (± 0.57)** |

methods[4], and `GPT-FL` under VGG19; and Figure 3 shows the communication cost comparison between generative data-based methods and `GPT-FL` under ConvNet. The target test accuracies in Figure 3 are set to be lower given the low accuracies achieved by FedGen. As shown, `GPT-FL` has the least communication cost among all the methods, achieving up to 94% communication reduction compared to the best-performed public data-based baseline Fed-ET and 98% communication reduction compared to the best-performed generated data-based baseline DynaFed. These results highlight the significant advantage of `GPT-FL` in communication reduction over state-of-the-art FL methods.

**Client Sampling Efficiency.** One critical hyper-parameter of FL is the client sampling rate in each communication round during the federated training process. In Figure 4, we plot the test model accuracies obtained by `GPT-FL` under low, medium, and high client sampling rates on CIFAR-10/100 with VGG19 under high data heterogeneity. As shown, even with a single participating client per round, `GPT-FL` is able to achieve 80.44% and 43.07% test accuracy on CIFAR-10 and CIFAR-100 respectively. This performance already surpasses all the other FL methods listed in Table 2, which employs 9 times more clients for training per round. These results highlight the significant advantage of `GPT-FL` in client sampling efficiency over state-of-the-art FL methods, making `GPT-FL` a very attractive solution in challenging scenarios where not many clients can participate at the same time.

### 4.2 UNDERSTANDING GPT-FL

**(1) Can we only rely on centralized training with synthetic data to achieve competitive results compared to Federated Learning with private data?**

To answer this question, we compare the model performance between generated downstream model by centralized training with synthetic data and the global model by standard FL training with private data on both image and audio benchmark datasets. Different from the previous section, we select ResNet18 and ResNet50 models for CIFAR-10 and CIFAR-100 dataset, respectively. We choose the models proposed in the FedAudio Benchmark Zhang et al. (2023) for audio tasks. We report the best F1 score for the audio datasets. The results are summarized in Table 3.

**Impact of Out-of-Domain Data Generation.** We choose the ESC-50 and Google Speech Commands datasets to examine the impact of out-of-domain data generation for the generative pre-trained model. We did not conduct a similar analysis for the image datasets as the LAION-5B Schuhmann et al. (2022) open-source dataset for training the Stable Diffusion model we used is a vast collection of publicly available datasets, including nearly all relevant ones for our experiments.

Our experiments show that synthetic image outperforms synthetic audio regarding model performance when using centralized training. We observed that centralized training with synthetic images achieves higher accuracy than FL setups for all three image benchmark datasets. In contrast, centralized training with synthetic audio performs worse than FL setups for ESC-50 and Google Speech Command datasets. The finding from the Google Speech Command experiments aligns with the previous study Li et al. (2018) that utilizes pure synthetic speech data to train the automatic speech recognition system leading to substantial performance degradation. One plausible explanation is related to the relatively small training data sizes (approximately 400M sentences) and constrained domain knowledge (book corpus) compared to training other generative pre-trained models like Stable Diffusion. For example, using human inspection, we discovered that the TTS model fails to synthesize simple spoken words like "house". This deficiency may originate from the lack of short-spoken utterance samples in training data. In addition, the synthesized speech often lacks diversity due to the limited range of speakers represented in the training dataset. On the other hand, there is insufficient knowledge of

---

[4]We do not compare with FedDF and DS-FL as they do not achieve competitive model accuracy.

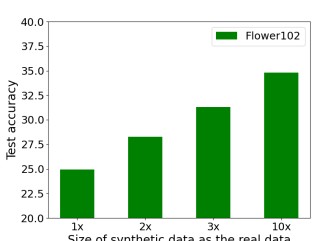
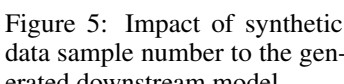
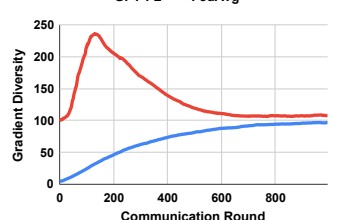
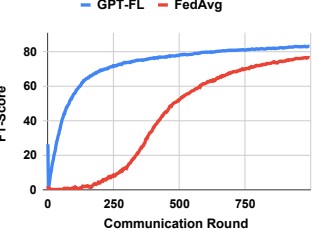

Figure 5: Impact of synthetic data sample number to the generated downstream model.

Figure 6: Smoothed Gradient diversity of client updates during training on Google speech commands dataset.

Figure 7: Learning curve of the global model during training on Google speech commands dataset.

the audio generation models, making the model performance of using the synthesized audio data as training data remains unknown. However, our manual inspection revealed that the model frequently encounters difficulties in generating audio samples, such as generated audio related to water sounds often sounds like music. This issue could be largely associated with the relatively small data size in pre-training compared to other foundation models.

**Impact of Numbers of Synthetic Data.** With both image and audio data, one commonality is centralized training with synthetic data can benefit from increasing the number of synthetic data. To validate this finding, we test the impact of numbers of synthetic data on the performance of the generated model on the Flowers102 dataset, where we increase the size of the synthetic data up to ten times that of the real data. As shown in Figure 5, our experimental results demonstrate that as we enlarge the amount of synthetic data, the performance of the model improves. One justification for this finding is that enlarging the number of synthetic data enriches the diversity and increases overlap between the synthetic and real data, allowing the model to learn more robust and generalizable features. Even if the data is generated randomly by the label name without any other diversity-enriching guidance from the real data, with more synthetic data, there is an increasing chance that some of these additional synthetic data overlap with the real data, allowing the model to perform better on the real test data.

**(2) What benefits does `GPT-FL` bring?**

We explore the benefits that `GPT-FL` provides for custom models that are built on top of downstream models generated from synthetic data. Specifically, we want to examine how fine-tuning these downstream models with private data under the FL framework can lead to performance improvements. To demonstrate how `GPT-FL` can be integrated with existing FL server optimizers, we evaluate its performance with both FedAvg and FedOpt as the server aggregator. Our experimental results are presented in Table 4. We also compare local fine-tuning in isolation with FL fine-tuning in Appendix.

Table 4: Accuracy comparison between generated downstream model, standard FL and `GPT-FL`. Differ from the experiments shown in Table 1, the CIFAR-10 and Flowers102 datasets are trained with ResNet18 model and the CIFAR-100 dataset is trained with ResNet50 model. "ΔMetric" represents the accuracy increment by `GPT-FL` on top of the generated downstream model.

| Dataset | 3x Synthetic | FedAvg | FedOpt | GPT-FL w/ FedAvg | GPT-FL w/ FedOpt | ΔMetric |
|---|---|---|---|---|---|---|
| CIFAR-10 | 75.65 (± 0.09) | 64.48 (± 0.13) | 72.68 (± 0.22) | **81.38 (± 0.05)** | 79.08 (± 0.17) | ↑ **5.73** |
| CIFAR-100 | 41.76 (± 0.03) | 25.89 (± 0.67) | 20.85 (± 0.14) | **62.83 (± 0.31)** | 48.80 (± 0.12) | ↑ **21.07** |
| Flowers102 | 31.29 (± 0.18) | 30.30 (± 0.16) | 26.43 (± 0.09) | 70.56 (± 0.34) | **77.57 (± 0.03)** | ↑ **46.28** |
| Google Command | 26.24 (± 0.01) | 73.68 (± 0.49) | 83.01 (± 0.23) | 81.90 (± 0.20) | **83.46 (± 0.11)** | ↑ **57.22** |
| ESC-50 | 12.72 (± 0.31) | 22.76 (± 1.01) | 32.49 (± 0.57) | 41.80 (± 0.32) | **43.46 (± 0.30)** | ↑ **30.74** |

**Effectiveness of Private Data.** Our experiments demonstrate the effectiveness of incorporating private data with FL into the finetuning process of the downstream model generated from synthetic data. As shown in Table 4, regardless of the modality and quality of the synthetic data used to generate the downstream model, FL fine-tuning leads to significant performance gains, outperforming the ones trained solely with FL or CL combined with synthetic training by a large margin. Furthermore, we observe that fine-tuning with private data can especially benefit the cases for out-of-domain synthetic data, such as in the audio data. For example, `GPT-FL` with FedOpt could achieve 43.46

test accuracy in the ECS-50 dataset, which nearly provides two times increment than standard FL and three times increment than centralized training by synthetic data. These results suggest that leveraging private data with FL in the fine-tuning process can greatly enhance the performance of synthetic data-generated models, making them more suitable for real-world applications.

**Generated Downstream Model Helps FL Optimization.** To gain a comprehensive understanding of why the custom models built using `GPT-FL` provide benefits to performance improvements, we decided to compare the gradient diversity between model weights initialized by `GPT-FL` and random initialization. Specifically, we apply the definition of the gradient diversity introduced from Yin et al. (2018) by adapting the gradients $g_i$ to client update $\Delta_i$:

$$\Delta_S = \frac{\sum_{i \in S} ||\Delta_i||^2}{||\sum_{i \in S} \Delta_i||^2} \tag{1}$$

where $S$ is the set of sampled clients in each communication round and $i$ represents the client index. As shown in Figure 6, the gradient diversity plot for FedAvg reveals that `GPT-FL` displays lower initial gradient diversity compared to random initialization. Over training time, both `GPT-FL` and random initialization converge to similar gradient diversity levels, consistent with the performance curve in Figure 7, where a larger $\Delta_S$ corresponds to slower convergence rate. This aligns with prior findings Nguyen et al. (2022), indicating that starting from a pre-trained model leads to less variation in local client updates, potentially addressing the client drift issue.

Table 5: Accuracy performance comparison between generated downstream model, standard federated learning and `GPT-FL`. All the training is initialized by ImageNet-based pre-train model.

| Dataset | 3x Synthetic | FedAvg | FedOpt | `GPT-FL` w/ FedAvg | `GPT-FL` w/ FedOpt |
|---------|-------------|--------|--------|-------------------|-------------------|
| CIFAR-10 | 72.65 (± 0.05) | 66.10 (± 0.03) | 79.08 (± 0.39) | 75.87 (± 0.73) | **82.20 (± 0.61)** |
| CIFAR-100 | 42.30 (± 0.01) | 62.83 (± 0.03) | 45.27 (± 0.10) | **66.84 (± 0.05)** | 66.03 (± 0.02) |
| Flowers102 | 41.05 (± 0.26) | 80.73 (± 0.01) | 87.33 (± 0.29) | 86.18 (± 0.04) | **88.66 (± 0.40)** |

**Harmonization With Existing Pre-train Model.** As the standard FL framework, `GPT-FL` could also benefit from other existing pre-train models. Specifically, besides training from scratch, `GPT-FL` could utilize the existing pre-train model to start training the synthetic data to generate downstream model and finetune it again with private data in FL. Table 5 presents the performance evaluation of `GPT-FL` on top of the pre-trained models for image datasets. We follow the approach from prior work Nguyen et al. (2022) and use the ImageNet pre-trained model available in the PyTorch Torchvision library. Our experiments show that `GPT-FL` achieves better results compared to training solely with FL or synthetic data, as reported in Table 4. Notably, the improvement in performance is consistent across three image benchmark datasets, with a gain ranging from 1% to 11% compared to the results in Table 4. These results demonstrate that `GPT-FL` can effectively leverage pre-trained models to improve performance in the FL setting.

## 5 CONCLUSION

We present `GPT-FL`, a generative pre-trained model-assisted federated learning framework. `GPT-FL` leverages the generative pre-trained model to generate diversified synthetic data for a wide range of data modalities before FL training. This synthetic data is then utilized to construct a downstream model, which undergoes fine-tuning with private data within a standard FL framework. Our experimental results showcase the remarkable performance of `GPT-FL` when compared to state-of-the-art FL methods. Moreover, through detailed ablation studies, we demonstrate that `GPT-FL` is a flexible and applicable framework solution to the challenges associated with cross-device FL scenarios.

**Limitations and Future works.** Due to the limitations in our computational resources, we cannot further scale up the synthetic data volume in our study, as it may take several weeks for the generation. Besides, we do not investigate the larger model sizes in our current study, which we will pursue it as our future work. In addition, we want to explore the expansions of the GPT-FL framework. GPT-FL seamlessly integrates with the vanilla FL framework, allowing for harmonization with most of the existing FL methods. We are interested in exploring the combination of the public-data-based FL aggregation scheme and the GPT-FL framework by replacing the public data with synthetic data.

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
