# A  APPENDIX

## A.1  INTEGRATION OF IBLT IN `GPT-FL`

Within the `GPT-FL` framework, the set of distinct label names is sourced from an open domain. The server lacks detailed length information on the set, making it challenging to directly encode the label names properly for secure aggregation. To address this, we propose to locally encode the unique label names into Invertible Bloom Lookup Tables (IBLT) Goodrich & Mitzenmacher (2011) data structure, a randomized data structure efficient in storing key-value pairs within an open domain. IBLT is a bloom filter-type linear data structure that supports the efficient listing of inserted elements and their precise counts, with table size scaling linearly with unique keys. IBLT sketches are amenable to linear summation, thus compatible with secure aggregation protocols.

In the `GPT-FL` framework's IBLT integration, each client locally encodes its distinct label names into IBLT and transmits it to the server. The server performs linear aggregation of these IBLTs through a secure multi-party computation protocol, subsequently decoding the aggregated table to obtain total label name counts without revealing individual label information. By leveraging the collective label name histogram, the server determines the union of distinct label names for data generation, maintaining the privacy of client-specific details. This approach finds validation in prior research Gascón et al. (2023), where IBLT demonstrated its efficacy in addressing private heavy hitters within federated analytics.

To better demonstrate the integration of IBLT in `GPT-FL`, we provide an illustrated experiment as an example. The experiment is conducted with the TensorFlow Federated IBLT API TensorFlow (2023). We partition the CIFAR-10 dataset heterogeneously amongst 100 clients using the Dirichlet distribution $Dir_K(\alpha)$ with $\alpha$ equal to 0.1. As the server does not know the length of the dataset initially, we set the capacity of the IBLT sketch to 50, which is much larger than the total number of unique labels inside CIFAR-10 (i.e., 10). Each client encodes its unique set of label names into IBLT and sends it to the server. The server would aggregate them via the secure aggregation protocol, which means the server can not access the individual IBLT but only knows the summation of IBLTs. After decoding the aggregated IBLT, the server only gets the following information:

```
Number of clients participated: 100
Discovered label names and counts:
{'dog': 49, 'automobile': 59, 'bird': 50, 'horse': 32, 'cat': 46,
    'frog': 27, 'deer': 44, 'truck': 37, 'airplane': 50, 'ship': 35}
```

The decode information only contains the number of participated clients and the histogram of the label name, which the server could infer the union of distinct label names for data generation. For example, the notation "'dog':49" denotes there are 49 clients who include the label 'dog' within their local datasets, but the server lacks knowledge regarding the specific client identities associated with this 'dog' label in the localized data. It is crucial to emphasize that the server remains unable to access specific client details, such as the labels held by individual clients. As suggested in the previous work TensorFlow (2023); Gascón et al. (2023), this algorithm could be further enhanced by adding a differential privacy mechanism. In conclusion, this IBLT-based algorithm will allow parties to jointly compute the union of unique label names without revealing individual label information, addressing concerns about privacy and confidentiality.

## A.2  EVALUATING CLIENT-ISOLATED FINE-TUNING AGAINST GPT-FL PERFORMANCE

To underscore the effectiveness of federation in fine-tuning, we present an ablation study comparing the performance of local fine-tuning in isolation against FL fine-tuning. This study utilizes the Google Speech Command and CIFAR-100 datasets. We partition the CIFAR-10 dataset using the Dirichlet distribution $Dir_K(\alpha)$ with $\alpha$ equal to 0.1 into 100 clients, and partition the Google Speech Command dataset over speaker IDs into 2,618 clients. We select the VGG19 model for CIFAR-100 dataset to align with Table 2. In the isolated fine-tuning scenario, we select 10 clients at random, allowing each to fine-tune the synthetic-data-based downstream model independently with its local data for 500 epochs. The average accuracy from these clients is then computed. For the GPT-FL setup, we maintain the experimental parameters as per the setups described in Table 4. The results of these experiments are summarized in Table 6.

Table 6: Accuracy performance comparison between locally fine-tune in isolation and `GPT-FL` with FedAvg.

| Dataset | Locally Fine-tune in Isolation | GPT-FL w/ FedAvg |
|---|---|---|
| CIFAR-100 | 35.53% | **47.80%** |
| Google Speech Command | 23.00% | **81.90%** |

The results show that fine-tuning in isolation at the client level yields significantly lower accuracy compared to the GPT-FL approach using FedAvg, which fine-tunes synthetic-data-based downstream model models federatively using private data. The primary reason for this disparity is the limited amount and skewed label distribution of the local data available to each client, which is insufficient for individually tuning the model to achieve optimal performance. These findings clearly demonstrate the value of federated learning in fine-tuning, especially given the limitations of local data in terms of volume and diversity.

### A.3 EVALUATION OF `GPT-FL` ON DOMAIN-SPECIFIC TASKS

In this section, we aim to comprehensively assess the effectiveness of the `GPT-FL` framework on domain-specific tasks that have limited overlap with the training data of the Stable Diffusion generator used in this study. All experiments in this section utilize the ResNet-18 model.

Two specific datasets, namely, COVID-19 X-rays Chowdhury et al. (2020) and Food101 Bossard et al. (2014), are chosen for evaluation in this section. The COVID-19 X-rays dataset is a publicly available collection of chest X-ray images with varying dimensions. It comprises three classes: COVID-19 X-ray images, normal X-ray images, and viral pneumonia X-ray images. As the database is updated randomly, we follow the previous FL work Kumar et al. (2022) and select the COVIDx-8A version in this paper. The dataset contains 5,585 training images and 400 test images. We partition the dataset into 20 clients using the Dirichlet distribution $Dir_K(\alpha)$ with $\alpha$ equal to 0.1. It's important to note that during prompt generation, we observed that the latest version of ChatGPT does not support content generation related to medical imagery. Consequently, we utilize label names as prompts to guide the generative model for synthetic X-ray image generation. As an illustration, one of the queries used to instruct ChatGPT to generate the prompt is as follows:

```
Q: Create a prompt to a chest x-ray with Viral Pneumonia
A: I'm unable to generate images of chest X-rays with viral
   pneumonia as it does not align with our content policy.
   This policy ensures the responsible and ethical use of AI,
   especially in sensitive areas like medical imagery. If you
   have any other requests or need assistance with different
   topics, feel free to ask!
```

The Food101 dataset contains 101 food categories with 101,000 images in total. Each class has 750 training images and 250 test images. The dataset is designed to contain some amount of noise in the training images, which comes mostly in the form of intense colors and sometimes wrong labels. All images were rescaled to have a maximum side length of 512 pixels. We partition the dataset into 50 clients using the Dirichlet distribution $Dir_K(\alpha)$ with $\alpha$ equal to 0.1. We use the same pipeline to generate the synthetic data for this dataset as we describe in Section 3.

Table 7: Accuracy performance comparison between generated downstream model, standard federated learning and `GPT-FL` on domain-specific tasks.

| Dataset | 3x Synthetic | FedAvg | GPT-FL w/ FedAvg |
|---|---|---|---|
| COVID-19 X-rays | 37.71% | 78.36% | **94.65%** |
| Food101 | 35.14% | 43.25% | **70.57%** |

The experiment results are shown in Table 7. In the experiments, we randomly sample 10 clients from 20 clients for the COVID-19 X-rays dataset and randomly sample 10 clients from 50 clients for the Food101 dataset. The results demonstrate that even though the synthetic data has significant differences compared to actual data, GPT-FL still provides performance benefits compared to exclusive reliance on FL, which aligns with our results in the main paper.

## A.4 EXPERIMENT SETTINGS

### A.4.1 COMPUTING INFRASTRUCTURE

All experiments are conducted via CPU/GPU simulation. The simulation experiments are performed on two computing servers with ten GPUs. The server is equipped with AMD EPYC 7502 32-Core Processor and 1024G memory. The GPU is NVIDIA RTX A100.

### A.4.2 DATASETS AND MODELS

**CIFAR-10.** The CIFAR-10 dataset Krizhevsky (2009) consists of 60,000 32x32 color images in 10 classes. It has 50,000 training images and 10,000 test images. We normalize the images using the mean and standard deviation of the dataset. For evaluation, we use ConvNet Pi et al. (2022), ResNet18 He et al. (2015), and VGG19 Simonyan & Zisserman (2014) models. Following the previous work Pi et al. (2022), the ConvNet has 3 layers with a hidden dimension of 128. The dataset is partitioned using a Dirichlet distribution to emulate a realistic non-iid distribution, following prior work Cho et al. (2022).

**CIFAR-100.** The CIFAR-100 dataset Krizhevsky (2009) is similar to CIFAR-10 but contains 100 classes, with 600 images per class. We apply the same partitioning method as CIFAR-10. For evaluation, we use ConvNet Pi et al. (2022), ResNet50 He et al. (2015), and VGG19 Simonyan & Zisserman (2014) models. The ConvNet architecture is the same as used for CIFAR-10.

**Oxford Flowers 102.** The Oxford Flowers 102 Nilsback & Zisserman (2008) (Flowers102) dataset consists of 102 types of flowers, with each type containing between 40 and 258 images. The images exhibit significant variations in scale, angle, and lighting. Some flower categories also have substantial variations within the category and contain several closely related categories. It is divided into training, validation, and test sets. The training and validation sets consist of 10 images per class, totaling 1020 images each. The test set contains the remaining 6149 images, with a minimum of 20 images per class. We resize all images to 224x224 pixels for consistency. For evaluation, we use ConvNet Pi et al. (2022), ResNet18 He et al. (2015), and VGG19 Simonyan & Zisserman (2014) models. We apply the same partitioning method as CIFAR-10. The ConvNet architecture is the same as used for CIFAR-10.

**Google Command.** The Google Command dataset Warden (2018) comprises 105,829 audio recordings collected from 2,618 speakers. The training set includes recordings from 2,112 speakers, the validation set includes 256 speakers, and the test set includes 250 speakers. It consists of 35 common words from everyday vocabulary, such as "Yes," "No," "Up," and "Down." For evaluation, we use a lightweight model based on related work Zhang et al. (2023) for a 35-class keyword spotting task, where the model consists of two convolution layers followed by one Gated Recurrent Units (GRU) layer and an average pooling layer is connected to the GRU output, which is then fed through two dense layers to generate the predictions. In this work, to pre-process the raw audio data, a sequence of overlapping Hamming windows is applied to the raw speech signal with a time shift of 10 ms. We calculate the discrete Fourier transform (DFT) with a frame length of 1,024 and compute the Mel-spectrogram with a dimension of 128. The Mel-spectrogram is used for training the keyword spotting model. We follow Zhang et al. (2023) for this setup.

**ESC-50.** The ESC-50 dataset Piczak consists of 2000 environmental audio recordings suitable for environmental sound classification. The dataset contains 5-second-long recordings categorized into 50 semantical classes, with 40 examples per class. These classes are loosely arranged into five major categories: animals, natural soundscapes & water sounds, human & non-speech sounds, interior/domestic sounds, and exterior/urban noises. We employ the same data pre-processing method and model architecture as used in the Google Command dataset.

### A.4.3 HYPERPARAMETER SETTINGS

To determine the optimal hyperparameters, we conducted a search within specified ranges. The client learning rate was searched in the range of 1.00E-09 to 1.00E-00, the server learning rate in the range of 1.00E-09 to 1.00E-00, weight decay in the range of 0.1 to 0.9, input batch size in the range of 8 to 256, and epoch number for centralized training in the range of 100 to 500. The hyperparameter settings for the public data-based methods and standard FL methods in Table 2 followed the settings from the previous work Cho et al. (2022). The specific hyperparameter selections for the other experiments are provided below.

**Hyperparameter Selection in Table 2.** The detailed experiment setups for Table 2 are listed in Table 8, Table 9, Table 10 and Table 11. For the experiments related to FedGen[5] and DynaFed[6], we evaluate them with their official implementation code on GitHub.

Table 8: Experimental setup details of `GPT-FL` with VGG19 in Table 2

|  |  | CIFAR-10 | CIFAR-100 | Flowers102 |
|---|---|---|---|---|
| Local Epoch |  | 1 | 1 | 1 |
| Communication Rounds |  | 500 | 500 | 500 |
| Cohort Size |  | 10 | 10 | 50 |
| Batch Size |  | 32 | 32 | 32 |
| Client Learning Rate | High Data Heterogeneity | 1.00E-07 | 1.00E-06 | 5.00E-03 |
|  | Low Data Heterogeneity | 1.00E-07 | 1.00E-06 | 5.00E-03 |
| Optimizer |  | SGD | SGD | SGD |
| Momentum |  | 0.9 | 0.9 | 0.9 |
| Weight Decay |  | 5.00E-04 | 5.00E-04 | 5.00E-04 |

Table 9: Experimental setup details of `GPT-FL` with ConvNet in Table 2

|  |  | CIFAR-10 | CIFAR-100 | Flowers102 |
|---|---|---|---|---|
| Local Epoch |  | 1 | 1 | 1 |
| Communication Rounds |  | 500 | 500 | 500 |
| Cohort Size |  | 10 | 10 | 50 |
| Batch Size |  | 32 | 32 | 32 |
| Client Learning Rate | High Data Heterogeneity | 2.00E-07 | 1.00E-04 | 1.00E-04 |
|  | Low Data Heterogeneity | 5.00E-06 | 1.00E-04 | 5.00E-03 |
| Optimizer |  | AdamW | AdamW | SGD |
| Betas |  | (0.9, 0.999) | (0.9, 0.999) | N/A |
| Eps |  | 1.00E-08 | 1.00E-08 | N/A |
| Weight Decay |  | 5.00E-04 | 5.00E-04 | 5.00E-04 |

Table 10: Experimental setup details of FedGen with ConvNet in Table 2

|  |  | CIFAR-10 | CIFAR-100 | Flowers102 |
|---|---|---|---|---|
| Local Epoch |  | 1 | 5 | 5 |
| Communication Rounds |  | 500 | 500 | 500 |
| Cohort Size |  | 10 | 10 | 50 |
| Batch Size |  | 32 | 32 | 32 |
| Generator Batch Size |  | 32 | 32 | 32 |
| Client Learning Rate | High Data Heterogeneity | 1.00E-02 | 1.00E-02 | 1.00E-02 |
|  | Low Data Heterogeneity | 1.00E-02 | 1.00E-02 | 1.00E-02 |
| Ensemble Learning Rate |  | 1.00E-04 | 1.00E-04 | 1.00E-04 |
| Personal Learning Rate |  | 1.00E-02 | 1.00E-02 | 1.00E-02 |
| Optimizer |  | Adam | Adam | Adam |
| Betas |  | (0.9, 0.999) | (0.9, 0.999) | (0.9, 0.999) |
| Eps |  | 1.00E-08 | 1.00E-08 | 1.00E-08 |
| Weight Decay |  | 1.00E-02 | 1.00E-02 | 1.00E-02 |

**Hyperparameter Selection in Table 3 and Table 4.** For the centralized training in Table 3 and Table 4, we used the following hyperparameter settings. For image data, the batch size was set to 32, and the optimizer was AdamW with weight decay set to 0.9 and cosine annealing learning rate decay.

---

[5]FedGen: https://github.com/zhuangdizhu/FedGen

[6]DynaFed: https://github.com/pipilurj/DynaFed/tree/main

Table 11: Experimental setup details of DynaFed with ConvNet in Table 2

|  |  | CIFAR-10 | CIFAR-100 | Flowers102 |
|---|---|---|---|---|
| Local Epoch | | 1 | 1 | 1 |
| Communication Rounds | | 500 | 500 | 500 |
| Cohort Size | | 10 | 10 | 50 |
| Batch Size | | 32 | 32 | 32 |
| Synthetic Images Learning Rate | | 5.00E-02 | 5.00E-02 | 5.00E-02 |
| Distill Interval | | 1 | 1 | 1 |
| Distill Iteration | | 20 | 8 | 20 |
| Distill Step | | 3000 | 200 | 500 |
| Distill Learning Rate | | 1.00E-04 | 1.00E-04 | 1.00E-04 |
| Client Learning Rate | High Data Heterogeneity | 1.00E-02 | 1.00E-02 | 1.00E-02 |
| | Low Data Heterogeneity | 1.00E-02 | 1.00E-02 | 1.00E-02 |
| Ensemble Learning Rate | | 1.00E-04 | 1.00E-04 | 1.00E-04 |
| Personal Learning Rate | | 1.00E-02 | 1.00E-02 | 1.00E-02 |
| Optimizer | | Adam | Adam | Adam |
| Betas | | (0.9, 0.999) | (0.9, 0.999) | (0.9, 0.999) |
| Eps | | 1.00E-08 | 1.00E-08 | 1.00E-08 |
| Weight Decay | | 1.00E-02 | 1.00E-02 | 1.00E-02 |

The initial learning rate was 1.00E-04 for CIFAR-10/CIFAR-100 and 3.00E-04 for Flowers102. For audio data, the batch size was set to 64, and the optimizer was Adam with weight decay set to 1.00E-04. The initial learning rate was 5.00E-05 for both datasets.

For the standard FL training in Table 3 and Table 4, the hyperparameter settings are as follows. For image data, the batch size is set to 32, and SGD is used as the local optimizer with weight decay set to 5.00E-04. When using FedOpt as the server aggregator, Adam is chosen as the server optimizer. Specifically, for the CIFAR-10 dataset, the local learning rate is set to 1.00E-01 with FedAvg as the server aggregator, and for FedOpt as the server aggregator, the local learning rate is set to 1.00E-02 and the server learning rate is set to 1.00E-03. For the CIFAR-100 dataset, the local learning rate is set to 1.00E-01 with FedAvg as the server aggregator, and for FedOpt as the server aggregator, both the local and server learning rates are set to 1.00E-01. For the Flowers102 dataset, the local learning rate is set to 1.00E-01 with FedAvg as the server aggregator, and for FedOpt as the server aggregator, the local learning rate is set to 1.00E-02 and the server learning rate is set to 1.00E-02. For all audio data, the experimental settings strictly follow the FedAudio benchmark Zhang et al. (2023).

For the `GPT-FL` training in Table 3 and Table 4, the hyperparameter settings are as follows. For image data, the batch size is set to 32, and SGD is used as the local optimizer with weight decay set to 5.00E-04. When using FedOpt as the server aggregator, Adam is chosen as the server optimizer. Specifically, for the CIFAR-10 dataset, the local learning rate is set to 5.00E-04 with FedAvg as the server aggregator, and for FedOpt as the server aggregator, the local learning rate is set to 3.00E-04 and the server learning rate is set to 7.00E-04. For the CIFAR-100 dataset, the local learning rate is set to 1.00E-04 with FedAvg as the server aggregator, and for FedOpt as the server aggregator, the local learning rate is set to 5.00E-04 and the server learning rate is set to 1.00E-03. For the Flowers102 dataset, the local learning rate is set to 5.00E-03 with FedAvg as the server aggregator, and for FedOpt as the server aggregator, the local learning rate is set to 1.00E-04 and the server learning rate is set to 1.00E-04. For audio data, the batch size is set to 16, and SGD is used as the local optimizer with weight decay set to 5.00E-04. When using FedOpt as the server aggregator, Adam is chosen as the server optimizer. We set the local learning rate to 5.00E-02 with FedAvg as the server aggregator, and for FedOpt as the server aggregator, the local learning rate is set to 1.00E-03 and the server learning rate is set to 5.00E-04 for both two datasets.

**Hyperparameter Selection in Table 5.** For the centralized training in Table 5, the hyperparameter selection is follows. For all image data, we set the batch size to 32, and choose AdamW Loshchilov & Hutter (2017) as the optimizer with weight decay equal to 0.9 and cosine annealing learning rate decay. For the CIFAR-10 dataset, we used an initial learning rate of 8.00E-06; for the CIFAR-100 dataset, we used an initial learning rate of 5.00E-06; for the Flowers102 dataset, we used an initial learning rate of 2.00E-05.

For the standard FL training in Table 5, we use the hyperparameter setting as follows. For all image data, we set the batch size to 32, and choose SGD as the local optimizer with weight decay equal to 5.00E-04. With FedOpt as the server aggregator, we choose Adam as the server optimizer. For the CIFAR-10 dataset, we choose the local learning rate as 1.00E-01 with FedAvg as the server

aggregator and choose the local learning rate as 1.00E-03 and the server learning rate as 1.00E-03 with FedOpt as the server aggregator. For the CIFAR-100 dataset, we choose the local learning rate as 1.00E-02 with FedAvg as the server aggregator and choose the local learning rate as 5.00E-03 and the server learning rate as 7.00E-03 with FedOpt as the server aggregator. For the Flowers102 dataset, we choose the local learning rate as 1.00E-02 with FedAvg as the server aggregator and choose the local learning rate as 1.00E-04 and the server learning rate as 5.00E-04 with FedOpt as the server aggregator.

For `GPT-FL` training in Table 5, we use the hyperparameter setting as follows. For all image data, we set the batch size to 32, and choose SGD as the local optimizer with weight decay equal to 5.00E-04. With FedOpt as the server aggregator, we choose Adam as the server optimizer. For CIFAR-10 dataset, we choose the local learning rate as 1.00E-07 with FedAvg as the server aggregator and choose the local learning rate as 1.00E-07 and the server learning rate as 1.00E-05 with FedOpt as the server aggregator. For CIFAR-100 dataset, we choose the local learning rate as 1.00E-04 with FedAvg as the server aggregator and choose the local learning rate as 1.00E-04 and the server learning rate as 1.00E-05 with FedOpt as server aggregator. For Flowers102 dataset, we choose the local learning rate as 1.00E-02 with FedAvg as the server aggregator and choose the local learning rate as 1.00E-04 and the server learning rate as 1.00E-04 with FedOpt as the server aggregator.

## A.5  QUALITY OF THE GENERATED SYNTHETIC DATA

As shown in Table 2, `GPT-FL` outperforms both generated data-based approaches FedGen and DynaFed significantly across all experimental conditions. One plausible reason for this could be associated with the quality of the generated synthetic data. Specifically, both FedGen and DynaFed rely on training MLP-based generator networks to ensemble user information in a data-free manner, where the lightweight generator may have limitations in generating high-fidelity data. The results of Flowers102 provide empirical evidence that such a lightweight generator has constrained capabilities in synthesizing image output on input images with larger sizes, making it challenging for the global model to converge. To illustrate this, Figure 8 and Figure 9 illustrate the synthetic images generated by `GPT-FL` and DynaFed, respectively. As shown, the learned generator of DynaFed fails to generate high-fidelity data as in `GPT-FL`.

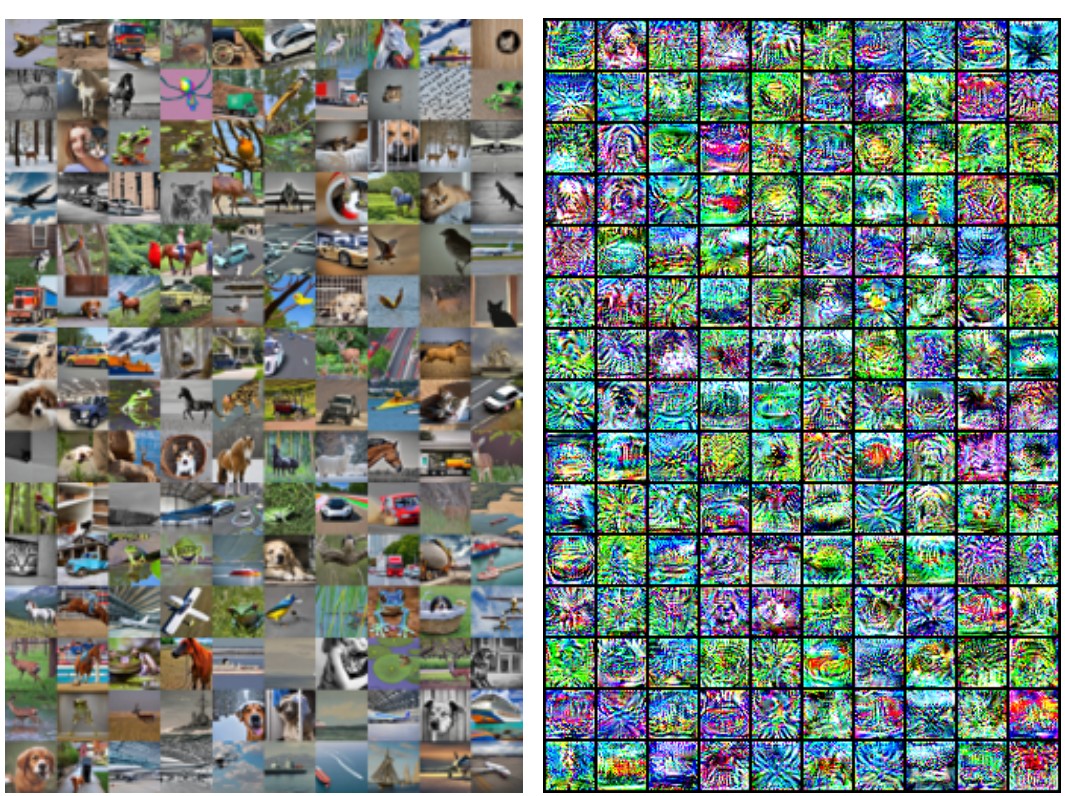

Figure 8: Synthetic CIFAR-10 data by `GPT-FL`.  Figure 9: Synthetic CIFAR-10 data by DynaFed.