# OpenReview forum: "GPT-FL: Generative Pre-trained Model-Assisted Federated Learning"
_ICLR.cc/2024/Conference — Submitted to ICLR 2024_

### Official Review · Reviewer_iSGe · 2023-10-29

**Soundness:** 3 good
**Presentation:** 3 good
**Contribution:** 3 good
**Rating:** 5
**Confidence:** 4

**Summary:**

This paper proposes a federated learning framework, which utilizes pre-trained generative models to synthesize data for server model training and then finetuning client models via local data. The proposed framework is validated on three datasets and shows higher performance than the compared methods.

**Strengths:**

- How to use synthetic data in FL is a relevant and important topic.
- The paper is well-organized and easy to follow.
- The authors analyzed the method from many aspects.

**Weaknesses:**

- The technical contribution is incremental; the framework mainly applies existing generative models.
- The methodology is agnostic to the FL framework. The generative-model-based pretraining can be done without FL. The specific relation with FL needs to be justified.
- It is not clear how to evaluate the quality of data generated by this framework.
- The motivation for why clients choose to finetune models in a federated way rather than locally is not justified. Better add the performance of performing local finetuning based on the GPT-pretrained server model.
- The comparison may not be proper. Some FL methods are agnostic to the pretraining. A better way would be to combine the GPT-FL and these methods to show the relative improvements.
- Fig.2 may not be fair. These methods start from scratch, and it is intuitive that these methods would require more communication to converge.
- The fourth finding that ‘downstream model generated by synthetic data controls gradient diversity during FL training’ is not new. This has been discussed by Nguyen et al.[1]

[1] Nguyen, John, Jianyu Wang, Kshitiz Malik, Maziar Sanjabi, and Michael Rabbat. "Where to begin? on the impact of pretraining and initialization in federated learning." ICLR, 2022.

**Questions:**

- Why does the MOON method involve the use of public data? The main idea of MOON is contrastive learning; the claim may not be correct and needs to be further justified.
- Why does FedOPT underperform 3x synthetic on image data but outperform it on audio data?
- Why change the model to ResNets in Table 4? Here needs to be further clarified.
- Using synthetic data seems promising. It would be interesting to further explore the effects of training larger models such as transformer-based methods.
- Does this method still work for more finer tasks? E.g., detection or segmentation.

---

> ### Author Response · Authors · 2023-11-17
> **Response to Reviewer iSGe (1/3)**
>
> Thank you for your time and suggestions. Here are our responses.
>
> ---
>
> ```
> Weakness 1: The technical contribution is incremental; the framework mainly applies existing generative models.
> ```
>
> **Response**: We believe our contributions are not incremental from three aspects.
> 1) Generated data-based FL is an emerging and very promising FL topic. At a high level, our work stands as one of the early and very few generated data-based FL works. More importantly, our work identifies the key limitations of state-of-the-art generated data-based FL approaches, and explains why such limitations are the fundamental reasons for their less competitive performance.
>
> 2) The design of our proposed framework, though seemingly straightforward, is not trivial at all. Our design decouples the synthetic data generation from the federated training process and leverages the knowledge from the generative pre-trained models, which directly addresses the limitations of state-of-the-art generated data-based FL approaches. Such design opens up new directions that no prior works explored, and pushes the state-of-the-arts forward.
>
> 3) Performance wise, our proposed approach not only outperforms generated data-based FL approaches, but also standard FL and public data-based FL approaches significantly, pushing the state-of-the-art FL performance forward.
>
> In the 4th paragraph of the introduction section, we summarize five key merits of our proposed approach over existing FL approaches. We also summarize the key empirical performance advantages in the last paragraph of the introduction section.
>
> ---
>
> ```
> Weakness 2: The methodology is agnostic to the FL framework. The generative-model-based pretraining can be done without FL. The specific relation with FL needs to be justified
> ```
>
> **Response**: We want to clarify that our framework is indeed a FL framework. The purpose of the centralized downstream model training (step 3 in Figure 1) is to provide a better initialized model for the following standard FL process (step 4 in Figure 1). This is the key difference from conventional FL where the model is usually randomly initialized.
>
> ---
>
> ```
> Weakness 3: It is not clear how to evaluate the quality of data generated by this framework.
> ```
>
> **Response**: We evaluated the quality of the generated data by following the well-recognized practices outlined in the latest work on measuring the quality of synthetic data from generative models [1]. Specifically, we evaluated the quality of the generated data by measuring the test accuracy of the model trained on the generated data but tested on real data. The accuracies are presented in Table 3, Table 4, and Table 5. This generated data quality evaluation method has also been adopted by the community in recent works [2,3]. The higher accuracy represents the closer the synthetic data is to the original data.
>
> [1] He, Ruifei, et al. "Is synthetic data from generative models ready for image recognition?." ICLR 2023.
>
> [2] Shipard, Jordan, et al. "Diversity is Definitely Needed: Improving Model-Agnostic Zero-shot Classification via Stable Diffusion." Proceedings of the IEEE/CVF Conference on Computer Vision and Pattern Recognition. 2023.
>
> [3] Azizi, Shekoofeh, et al. "Synthetic data from diffusion models improves imagenet classification.
>
> ---
>
> ```
> Weakness 4: The motivation for why clients choose to finetune models in a federated way rather than locally is not justified. Better add the performance of performing local finetuning based on the GPT-pretrained server model.
> ```
>
> **Response**: Based on your suggestion, we conducted an ablation study comparing local fine-tuning against FL fine-tuning with FedAvg. We did the experiments with Google Command and the Cifar100 dataset. For each experiment in local fine-tuning, we randomly select 10 clients to fine-tune locally in isolation and report the average accuracy. We partitioned the datasets the same as in the paper. More details about these experiments are shown in the revised version of the Appendix. The results are as follows:
>
> |                | Locally Fine-tune | FL Fine-tune with FedAvg |
> |----------------|-------------------|--------------------------|
> | CIFAR-100      | 35.53%            | 47.80%                   |
> | Google Command | 23.00%            | 81.90%                   |
>
> These findings clearly demonstrate the value of federated learning in fine-tuning, especially given the limitations of local data in terms of volume and diversity.

---

> ### Author Response · Authors · 2023-11-17
> **Response to Reviewer iSGe (2/3)**
>
> ```
> Weakness 5: The comparison may not be proper. Some FL methods are agnostic to the pretraining. A better way would be to combine the GPT-FL and these methods to show the relative improvements. Fig.2 may not be fair. These methods start from scratch, and it is intuitive that these methods would require more communication to converge.
> ```
>
> **Response**: We must emphasize that the results of GPT-FL in Table 2 are based on random initialization, the same as other methods in Table 2. As a result, it is a fair and proper comparison between GPT-FL and other baselines in Table 2.
>
> In fact, we indeed did experiments regarding GPT-FL’s harmonization with the existing pre-train model in the last part of Section 4.2. Specifically, besides training from scratch, GPT-FL
> could utilize the existing pre-train model to start training the synthetic data to generate a downstream model and finetune it again with private data in FL. As shown in Table 5, GPT-FL outperforms FL methods starting from a pre-trained model by a large margin.
>
> ---
>
> ```
> Weakness 6: The fourth finding that ‘downstream model generated by synthetic data controls gradient diversity during FL training’ is not new. This has been discussed by Nguyen et al.[1]
> ```
>
> **Response**: In Section 4.2, we indeed cite and discuss Nguyen et al.'s work in our paper to provide context for our own findings. Our contribution extends beyond the scope of Nguyen et al.’s research. While Nguyen et al. focused on the impact of starting federated learning from a pre-trained model, our work specifically examines the effects of using a downstream model generated by synthetic data. Our findings reveal that such models can similarly control gradient diversity during FL training, which aligns with Nguyen’s finding.

---

> ### Author Response · Authors · 2023-11-17
> **Response to Reviewer iSGe (3/3)**
>
> ```
> Question 1: Why does the MOON method involve the use of public data? The main idea of MOON is contrastive learning; the claim may not be correct and needs to be further justified.
> ```
>
> **Response**: We thank the reviewer for their insightful comment regarding our reference to the MOON method and the use of public data. Upon re-evaluation, we agree with your observation. In our initial manuscript, we incorrectly stated that MOON requires public data, following the assertions made in the Fed-ET paper [1]. We realize now that this interpretation was not accurate. We have revised the relevant sections in our paper.
>
> [1] Cho, Y.J., Manoel, A., Joshi, G., Sim, R., & Dimitriadis, D. Heterogeneous Ensemble Knowledge Transfer for Training Large Models in Federated Learning. IJCAI 2022.
>
> ---
>
> ```
> Question 2: Why does FedOPT underperform 3x synthetic on image data but outperform it on audio data?
> ```
>
> **Response**: This is because the quality of synthetic audio data is worse than that of synthetic image data. The experiment results with synthetic audio align with the previous study [1] that utilizes pure synthetic speech data to train the automatic speech recognition system leading to substantial performance degradation. We have thoroughly discussed it in the paper, and more details are stated in the third paragraph of Section 4.2.
>
> [1] Li, J., Gadde, R.T., Ginsburg, B., & Lavrukhin, V. (2018). Training Neural Speech Recognition Systems with Synthetic Speech Augmentation. ArXiv, abs/1811.00707.
>
> ---
>
> ```
> Question 3: Why change the model to ResNets in Table 4? Here needs to be further clarified.
> ```
>
> **Response**: In our experiments in Table 2, we opted for VGG19 and ConvNet to maintain consistency with the experimental setups of prior works [1,2] in the field. This choice was intended to facilitate direct comparison and benchmarking.
>
> However, we recognize that VGG19, being a relatively large model, may not be ideally suited for cross-device FL scenarios due to its computational and memory demands. In response to this consideration, we chose to employ ResNet models for the experiments presented in Table 4. ResNet models are not only lighter but also more commonly used in cross-device FL research. This makes them a more practical choice for such settings. The shift to ResNets also aligns with our objective to demonstrate the versatility of our GPT-FL framework in supporting various model architectures effectively.
>
> [1] Cho, Y.J., Manoel, A., Joshi, G., Sim, R., & Dimitriadis, D. Heterogeneous Ensemble Knowledge Transfer for Training Large Models in Federated Learning. IJCAI 2022.
>
> [2] Pi, R., Zhang, W., Xie, Y., Gao, J., Wang, X., Kim, S., & Chen, Q. (2022). DYNAFED: Tackling Client Data Heterogeneity with Global Dynamics. CVPR 2023.
>
> ---
>
> ```
> Question 4: Using synthetic data seems promising. It would be interesting to further explore the effects of training larger models such as transformer-based methods.
> ```
>
> **Response**: Thank you for your insightful suggestions regarding the exploration of larger models. Inspired by your recommendation, we initiated an experiment using the Vision Transformer (ViT) model to further validate our approach. However, due to the extensive computational resources and time required for training such large models, we were unable to complete this experiment within the rebuttal period.
>
> We plan to continue this line of investigation and will include the results in our paper as soon as they are available. This endeavor not only aligns with your suggestion but also opens up new avenues for future research. We recognize the potential of exploring various model architectures in our study and intend to delve deeper into this area. We believe that this will not only enrich our current work but also inspire further research within the community.
>
> ---
>
> ```
> Question 5: Does this method still work for more finer tasks? E.g., detection or segmentation.
> ```
>
> **Response**: Based on your suggestion, we have also added two new experiments regarding more domain-specific tasks with Food 101[1] and the COVID-19 X-rays[2] dataset in the revised Appendix.
>
> |                 | 3x Synthetic | FedAvg | GPT-FL w/ FedAvg |
> |-----------------|--------------|--------|------------------|
> | COVID-19 X-rays | 37.71%       | 78.36% | 94.65%           |
> | Food 101        | 35.14%       | 43.25% | 70.57%           |
>
> The results demonstrate that even though the synthetic data has significant differences compared to actual data, GPT-FL still provides performance benefits compared to exclusive reliance on FL.
>
> [1]. Bossard, Lukas et al. “Food-101 - Mining Discriminative Components with Random Forests.” ECCV 2014.
>
> [2]. Chowdhury, Muhammad Enamul Hoque et al. “Can AI Help in Screening Viral and COVID-19 Pneumonia?” IEEE Access 8 (2020): 132665-132676.
>
> ---
>
> If these responses have satisfactorily addressed your concerns, please consider increasing your score. Thank you!

---

> ### Author Response · Authors · 2023-11-19
>
> Dear Reviewer, we want to thank you again for your valuable comments. Feel free to let us know if you have any questions on our response.

---

> > ### Comment · Reviewer_iSGe · 2023-11-22
> > **Thanks for the rebuttal**
> >
> > Thanks for the rebuttal. I still have some concerns and questions.
> >
> > Weakness 1: The clarification is not convincing. The framework is new, but the technical components inside the framework are not new, e.g., generating prompts from label names, using existing pre-trained generative models.
> >
> > Weakness 2: In this case, this paper should present and discuss more convergence-related results.
> >
> > Weakness 3: Are there any other metrics to evaluate the data quality itself rather than testing accuracy?
> >
> > Weakness 4: The results look promising. Is the dataset non-iid distributed? What are the results if the clients have all class categories?
> >
> > Weakness 5: The third step of GPT-FL is to train the model on synthesized data. I mean, what if other methods also start from the third step of GTP-FL? In other words, can other methods benefit from the data synthesized by GPT-FL?

---

> ### Author Response · Authors · 2023-11-23
>
> Thank you for your reply. Here are our responses:
>
> ---
>
> ```
> Weakness 1: The clarification is not convincing. The framework is new, but the technical components inside the framework are not new, e.g., generating prompts from label names, using existing pre-trained generative models.
> ```
>
> The contributions to novelty can come from each technical component, but can also come from building such a new framework itself. More importantly, our new framework outperforms existing state-of-the-art FL frameworks by a large margin. This result is a solid proof of the novel contributions of our framework to the existing FL literature.
>
> ---
>
> ```
> Weakness 2: In this case, this paper should present and discuss more convergence-related results.
> ```
>
> In terms of convergence-related results, we show in Figure 6 and 7 that our proposed GPT-FL is able to significantly reduce the convergence time compared to state-of-the-art FL frameworks. Moreover, on page 9 of our paper, we followed [1] to perform a detailed analysis on gradient diversity. We reported that GPT-FL is able to achieve a lower gradient diversity compared to FedAvg,  which contributes to faster convergence.
>
> [1] Nguyen, John et al. “Where to Begin? On the Impact of Pre-Training and Initialization in Federated Learning.” ICLR 2023
>
> ---
>
> ```
> Weakness 3: Are there any other metrics to evaluate the data quality itself rather than testing accuracy?
> ```
>
> As described on page 7 of our paper, we employ a similar internal human inspection procedure to assess the quality of synthetic data, following the methodology used in a leading foundational model work [1]. We will add this point in our revised version.
>
>
> In addition, we want to emphasize that the performance of GPT-FL is not solely dependent on the quality of synthetic data. This is evidenced in Tables 4 and 5 of our original submission, as well as the new results from medical images and Food101 added during the rebuttal. These findings collectively illustrate that regardless of the modality and quality of the synthetic data, GPT-FL is able to effectively utilize the client data in FL, consistently achieving substantial performance improvements and faster convergence.
>
>
> [1] Liu, Haohe et al. “AudioLDM: Text-to-Audio Generation with Latent Diffusion Models.” ICML 2023.
>
> ---
>
> ```
> Weakness 4: The results look promising. Is the dataset non-iid distributed? What are the results if the clients have all class categories?
> ```
>
> The dataset is indeed non-iid distributed. We describe the details in the appendix of the revised version.
>
> For your new suggestion, we conducted another experiment on CIFAR-100 with iid data partition (Dirichlet distribution with alpha equal to 1.0). We could not modify the distribution of Google Command as it is naturally partitioned. We selected 10 clients, each with a complete set of class categories, and performed local fine-tuning in isolation. The results are as follows:
>
> |           | Locally Fine-tune (alpha = 0.1) | Locally Fine-tune (alpha = 1.0) | FL Fine-tune with FedAvg (alpha = 0.1) |   |
> |-----------|---------------------------------|---------------------------------|----------------------------------------|---|
> | CIFAR-100 | 35.53%                          | 39.64%                          | 47.80%                                  |   |
>
> These findings indicate that local fine-tuning, regardless of data distribution, struggles due to the limited data volume at each client, which is insufficient for achieving optimal performance, aligning with our previous statement.
>
> ---
>
> ```
> Weakness 5: The third step of GPT-FL is to train the model on synthesized data. I mean, what if other methods also start from the third step of GTP-FL? In other words, can other methods benefit from the data synthesized by GPT-FL?
> ```
>
> GPT-FL opens the door of a new FL paradigm that leverages synthesized data to enhance the performance of federated training. Given the promising results we obtained, we believe other FL methods can leverage GPT-FL to further enhance their own performance.
>
> ---
>
> We hope your concerns could be resolved and the rating of the paper can be increased accordingly. Thank you!

---

### Official Review · Reviewer_4UJh · 2023-10-30

**Soundness:** 3 good
**Presentation:** 4 excellent
**Contribution:** 3 good
**Rating:** 6
**Confidence:** 4

**Summary:**

This paper proposes a new approach for Federated Learning involving the use of synthetic data to assist model training. In particular, the proposed method uses the GPT model to generate image/audio prompts given label description, and then passes the prompt to text-to-image/text-to-audio generative models to create synthetic data. This dataset is then used to pre-train a downstream model that acts as an initializer to the FL training process. The paper shows empirical evidence that this improves FL performances.

**Strengths:**

I think this is a very interesting and novel approach to generally augment predictive models (not just in FL context).

The paper is clearly written and is very well motivated. Experiments are well thought out, thought provoking and yield promising results.

**Weaknesses:**

Regarding soundness, the motivation and the proposed method are very straight forward. I do not have any problem with the technical approach.

However, I think there should be some extra empirical study to better understand when the proposed method will and will not work. Below are several suggestions, which are not necessarily the weaknesses of this paper (although would be interesting if addressed/investigated).

GPT-FL seems to do very well on standard vision benchmark. How do we know that the generative model was not trained on these benchmark previously? Is there a chance that the StableDiffusion model has ingested enough test data and thus will provide an advantage to the downstream model?

I would suggest measuring the Fréchet Inception Distance to the training/test images (or some similar generative model evaluation metrics) to get a better understanding. I am not sure how to do that with audio data, but my hypothesis is that the closer the synthetic data to the training/test distribution, the better GPT-FL will perform -- which could explain Table 3.

Another ablation study that the authors can perform is to conduct FL training on tasks that are more domain specific (e.g., medical images) and hence less likely to overlap with the training data of the respective generative models.

The authors have conducted ablation studies to demonstrate two key points: (1) Centralized training with synthetic data alone cannot completely replace FL, which benefits from private siloed data (Table 3, downstream model vs. standard FL); (2) Synthetic data improves FL performance (Table 4, standard FL vs. FL initialized by downstream model). Those are important points, but I think it would also be interesting to investigate if each client can just fine-tune the downstream model in isolation and achieve the same level of performance with GPT-FL (i.e., local learning initialized by downstream model vs. FL initialized by downstream model). If this is the case, then there is no need to frame this contribution in the FL setting. It would instead be a new approach to synthesize a pre-trained model without data.

I find this paper which has a similar idea "PS-FedGAN: An Efficient Federated Learning Framework Based on Partially Shared Generative Adversarial Networks For Data Privacy" (Wijesinghe et al., 2023). The main difference is that they do not assume having a generative model at their disposal, and instead try to learn one. Do you foresee this would work better in the case where the FL task data are OOD of the pretrained generative model?

**Questions:**

I have put all my concerns above in question form.

---

> ### Author Response · Authors · 2023-11-17
> **Response to Reviewer 4UJh (1/2)**
>
> Thank you for your time and suggestions. Here are our responses.
>
> ---
> ```
> Weakness 1: GPT-FL seems to do very well on standard vision benchmark. How do we know that the generative model was not trained on these benchmark previously? Is there a chance that the StableDiffusion model has ingested enough test data and thus will provide an advantage to the downstream model?
>
> I would suggest measuring the Fréchet Inception Distance to the training/test images (or some similar generative model evaluation metrics) to get a better understanding. I am not sure how to do that with audio data, but my hypothesis is that the closer the synthetic data to the training/test distribution, the better GPT-FL will perform -- which could explain Table 3.
> ```
>
> **Response**: We evaluated the quality of the generated data by following the well-recognized practices outlined in the latest work on measuring the quality of synthetic data from generative models [1]. Specifically, we evaluated the quality of the generated data by measuring the test accuracy of the model trained on the generated data but tested on real data. The accuracies are presented in Table 3, Table 4, and Table 5. This generated data quality evaluation method has also been adopted by the community in recent works [2,3]. The higher accuracy represents the closer the synthetic data is to the original data.
>
> We have taken into account the impact of out-of-domain data generation, as discussed at the beginning of Section 4.2. We evaluated the synthetic audio data generated by SpeechT5 [4] and AudioLDM [5] models. The downstream task dataset, Google Command, does not overlap (e.g. speakers or utterances) with the LibriSpeech dataset [6], which is used to train the SpeechT5 model. The experience results in Table 3 show that centralized training with synthetic audio performs worse than FL.
>
> However, it is worth noting that GPT-FL does not solely rely on the quality of synthetic data. As highlighted in the benefits of GPT-FL and demonstrated in Table 4, the FL fine-tuning step consistently leads to significant performance gains, surpassing models trained solely with FL or FL combined with synthetic training by a large margin. These results emphasize that regardless of whether the target data falls within or outside the domain of the pre-trained generative model, GPT-FL can efficiently utilize the client data in FL, consistently achieving substantial performance improvements and faster convergence, surpassing the results obtained by models trained solely with real or synthetic data respectively.
>
> [1] He, Ruifei, et al. "Is synthetic data from generative models ready for image recognition?." ICLR 2023.
>
> [2] Shipard, Jordan, et al. "Diversity is Definitely Needed: Improving Model-Agnostic Zero-shot Classification via Stable Diffusion."
> Proceedings of the IEEE/CVF Conference on Computer Vision and Pattern Recognition. 2023.
>
> [3] Azizi, Shekoofeh, et al. "Synthetic data from diffusion models improves imagenet classification.
>
> [4] Ao, Junyi et al. “SpeechT5: Unified-Modal Encoder-Decoder Pre-Training for Spoken Language Processing.” ACL 2021.
>
> [5] Liu, Haohe et al. “AudioLDM: Text-to-Audio Generation with Latent Diffusion Models.” ICML 2023.
>
> [6] Panayotov, Vassil et al. “Librispeech: An ASR corpus based on public domain audio books.” ICASSP 2015
>
> ---
>
> ```
> Weakness 2: Another ablation study that the authors can perform is to conduct FL training on tasks that are more domain specific (e.g., medical images) and hence less likely to overlap with the training data of the respective generative models.
> ```
>
> **Response**: Based on your suggestion, we have also added two new experiments regarding more domain-specific tasks with Food 101[1] and the COVID-19 X-rays[2] dataset in the revised Appendix.
>
> |                 | 3x Synthetic | FedAvg | GPT-FL w/ FedAvg |
> |-----------------|--------------|--------|------------------|
> | COVID-19 X-rays | 37.71%       | 78.36% | 94.65%           |
> | Food 101        | 35.14%       | 43.25% | 70.57%           |
>
>
> The results demonstrate that even though the synthetic data has significant differences compared to actual data, GPT-FL still provides performance benefits compared to exclusive reliance on FL or synthetic data.
>
> [1]. Bossard, Lukas et al. “Food-101 - Mining Discriminative Components with Random Forests.” European Conference on Computer Vision (2014).
>
> [2]. Chowdhury, Muhammad Enamul Hoque et al. “Can AI Help in Screening Viral and COVID-19 Pneumonia?” IEEE Access 8 (2020): 132665-132676.

---

> > ### Comment · Reviewer_4UJh · 2023-11-20
> > **Discussion**
> >
> > First, I would like to thank the authors for taking the time to address my comments as well as providing new results. Regarding your responses:
> >
> > 1. I don't see how the first part of your answer to weakness 1 (i.e., referencing Table 3-5 in your manuscript) is relevant to the question regarding similarity between the training set of StableDiffusion and test images. The focus isn't about generating synthetic images with high quality. I just want to know if the pretrained model provides an unfair advantage if it was previously trained on the test images. However, I am somewhat satisfied with the second half of the answer, and also the answer to weakness 2.
> >
> > 2. Regarding your answer to weakness 3, what setting of GPT-FL was used to generate the new results? In particular, I'm not clear which column of Table 3 does "FL Fine-tune with FedAvg" correspond to. Will increasing the amount of synthetic data close the gap between fine-tuning in isolation and federated fine-tuning?

---

> > > ### Author Response · Authors · 2023-11-21
> > >
> > > Thank you for the reply. Below is our response:
> > >
> > > ```
> > > I don't see how the first part of your answer to weakness 1 (i.e., referencing Table 3-5 in your manuscript) is relevant to the question regarding similarity between the training set of StableDiffusion and test images. The focus isn't about generating synthetic images with high quality. I just want to know if the pretrained model provides an unfair advantage if it was previously trained on the test images. However, I am somewhat satisfied with the second half of the answer, and also the answer to weakness 2.
> > > ```
> > >
> > > **Response:** We apologize for the confusion. Just to be clear: the pretrained model was **NOT** previously trained on the test images.
> > >
> > > ```
> > > Regarding your answer to weakness 3, what setting of GPT-FL was used to generate the new results? In particular, I'm not clear which column of Table 3 does "FL Fine-tune with FedAvg" correspond to. Will increasing the amount of synthetic data close the gap between fine-tuning in isolation and federated fine-tuning?
> > > ```
> > >
> > > **Response:** “FL Fine-tune with FedAvg” corresponds to the column “GPT-FL w/ FedAvg” in Table 4. We use the same setting of GPT-FL as in Table 1 for CIFAR-100 and the same as in Table 4 for Google Command in the new results.
> > >
> > > Increasing the amount of synthetic data does **NOT** close the gap between fine-tuning in isolation and federated fine-tuning. We do an ablation study with CIFAR100 as shown below. The local model is VGG19 as we selected in Table 1.
> > >
> > > |                                                             | 1x Synthetic | 2x Synthetic | 3x Synthetic |
> > > |-------------------------------------------------------------|--------------|--------------|--------------|
> > > | Local learning initialized by downstream model              | 16.90%       | 27.89%       | 35.53%       |
> > > | FL initialized by downstream model (i.e., GPT-FL w/ FedAvg) | 35.61%       | 40.17%       | 47.80%       |
> > >
> > > The findings reveal that with one, two, and three times the amount of synthetic data compared to real data, the performance gap between isolated fine-tuning and federated fine-tuning is 18.71%, 12.28%, and 12.27%, respectively.
> > >
> > > While increasing the synthetic data does enhance the quality of the downstream model, both fine-tuning methods start with the same model as their initial point. Therefore, the increased synthetic data volume does not differentially affect these two approaches. The key factor contributing to the performance gap is the limited and skew-distributed nature of the local data. This inherent limitation in local data quality and distribution leads to poorer performance in isolated local fine-tuning, despite the improved quality of the initial downstream model from increased synthetic data.

---

> > > > ### Comment · Reviewer_4UJh · 2023-11-23
> > > > **Thanks for your responses**
> > > >
> > > > Dear authors,
> > > >
> > > > I appreciate the new results and I am happy with the explanation. Although I will keep my score for now (which already reflects my positive sentiment towards your work), I will consider raising it after discussing with other reviewers to see if their concerns have been addressed.

---

> > > > > ### Author Response · Authors · 2023-11-23
> > > > >
> > > > > Thank you very much.

---

> ### Author Response · Authors · 2023-11-17
> **Response to Reviewer 4UJh (2/2)**
>
> ```
> Weakness 3: I think it would also be interesting to investigate if each client can just fine-tune the downstream model in isolation and achieve the same level of performance with GPT-FL (i.e., local learning initialized by downstream model vs. FL initialized by downstream model). If this is the case, then there is no need to frame this contribution in the FL setting. It would instead be a new approach to synthesize a pre-trained model without data.
> ```
>
> **Response**: Following your suggestion, we conducted an ablation study comparing local fine-tuning against FL fine-tuning with FedAvg. We did the experiments using Google Command and Cifar100 datasets. For each experiment in local fine-tuning, we randomly select 10 clients to fine-tune locally in isolation and report the average accuracy. We partitioned the datasets the same as in the paper. More details about these experiments are shown in the revised version of the Appendix. The results are as follows:
>
> |                | Locally Fine-tune | FL Fine-tune with FedAvg |
> |----------------|-------------------|--------------------------|
> | CIFAR-100      | 35.53%            | 47.80%                   |
> | Google Command | 23.00%            | 81.90%                   |
>
> These findings clearly demonstrate the value of federated learning in fine-tuning, especially given the limitations of local data in terms of volume and diversity.
>
> ---
>
> ```
> Weakness 4: I find this paper which has a similar idea "PS-FedGAN: An Efficient Federated Learning Framework Based on Partially Shared Generative Adversarial Networks For Data Privacy" (Wijesinghe et al., 2023). The main difference is that they do not assume having a generative model at their disposal, and instead try to learn one. Do you foresee this would work better in the case where the FL task data are OOD of the pretrained generative model?
> ```
>
> **Response**: PS-FedGAN proposed to train a GAN-based generator at the server to cope with non-IID data distribution in FL training. As noted in our Introduction, one primary challenge with training generative models concurrently with federated learning is the instability of synthetic data quality during the early stages of convergence. This instability can adversely affect the FL training process. Our concern, as supported by evidence in Table 2, is that such unstable synthetic data, especially in the context of high-resolution imagery, would in turn jeopardize the federated training process. In PS-FedGAN, the absence of consideration for high-resolution images raises questions about its capability in high-resolution image generation. Our observations of related generator-based FL methods, specifically FedGen and DynaFed, indicate difficulties in converging on high-resolution datasets like Flower102. Considering these factors, we infer that our GPT-FL framework would likely offer superior performance compared to PS-FedGAN when dealing with OOD tasks. This inference is also based on the results of the audio datasets and the new-added ablation studies with the Food101 and Covid X-rays dataset.
>
> We initially wanted to do an experiment with PS-FedGAN for a direct comparison, but the paper does not provide the open-source code, which makes it difficult to reproduce within the rebuttal period. We have included PS-FedGAN in the related work section and marked it as a baseline for future studies.
>
> ---
>
> If these responses have satisfactorily addressed your concerns, please consider increasing your score. Thank you!

---

> ### Author Response · Authors · 2023-11-19
>
> Dear Reviewer, we want to thank you again for your valuable comments. Feel free to let us know if you have any questions on our response.

---

### Official Review · Reviewer_p1Ep · 2023-10-31

**Soundness:** 2 fair
**Presentation:** 3 good
**Contribution:** 1 poor
**Rating:** 3
**Confidence:** 4

**Summary:**

The present paper proposes a learning algorithm using generative model to enable the proposed algorithm to cope with heterogenous data distribution among clients. Specifically, the paper proposes that a central server employs a generative model to generate synthetic data samples. Then the synthetic data samples are used to train a global model at the server. Then the server sends the global model to clients and clients fine-tune the global model using their local data to obtain their personalized local models. The paper conducts extensive experiments on various tasks.

**Strengths:**

The paper conducts extensive experiments to evaluate the performance of the proposed algorithm.

**Weaknesses:**

I do not agree with authors that the proposed framework is a federated learning method. To train the global model at the server, the server does not need the collaboration of clients. In fact, participation of a client does not provide any benefit for others. In this case, federation is not needed. Therefore, clients themselves can use a generative model to generate synthetic data and then train a model using the synthetic data locally. After that client can fine-tune the model. The only reason that the server can be used for this purpose is that the server might have superior ability in training. To sum up, I do not think that the proposed framework makes a significant contribution. Also, the proposed algorithm do no need to train models iteratively and just one iteration should be enough.

Second, unlike the claim in the paper, I do not believe that the proposed framework can deal with client drift effectively. If the distribution of synthetic data is significantly different from those of clients, then fine-tuning the global model might be the same as training a model from scratch locally by the clients.

**Questions:**

No question.

---

> ### Author Response · Authors · 2023-11-17
> **Response to Reviewer p1Ep (1/2)**
>
> Thank you for your time and suggestions. Here are our responses.
>
> ---
> ```
> Weakness 1: I do not agree with authors that the proposed framework is a federated learning method. To train the global model at the server, the server does not need the collaboration of clients. In fact, participation of a client does not provide any benefit for others. In this case, federation is not needed. Therefore, clients themselves can use a generative model to generate synthetic data and then train a model using the synthetic data locally. After that client can fine-tune the model. The only reason that the server can be used for this purpose is that the server might have superior ability in training. To sum up, I do not think that the proposed framework makes a significant contribution. Also, the proposed algorithm do no need to train models iteratively and just one iteration should be enough.
> ```
>
> **Response**: We want to clarify that GPT-FL is indeed a FL framework but is different from the standard FL framework. The only difference between GPT-FL and standard FL framework is that in standard FL, the model is first randomly initialized and then distributed to the clients for federated training; in GPT-FL, the initial model is a pre-trained model from GPT. As shown in Figure 1 step 4, once we obtain this initial model, it will then be distributed to the clients for federated training, which is the same as the standard FL and needs the collaboration of the clients.
>
> As emphasized in the benefits of GPT-FL and demonstrated in Table 4, consistent results highlighted the significant performance gains from FL fine-tuning. To address your concern, we conducted an ablation study comparing local fine-tuning against FL fine-tuning with FedAvg. We did the experiments with Google Command and the Cifar100 dataset. For each experiment in local fine-tuning, we randomly select 10 clients to fine-tune locally in isolation and report the average accuracy. We partitioned the datasets the same as in the paper. More details about these experiments are shown in the revised version of the Appendix. The results are as follows:
>
> |                | Locally Fine-tune | FL Fine-tune with FedAvg |
> |----------------|-------------------|--------------------------|
> | CIFAR-100      | 35.53%            | 47.80%                   |
> | Google Command | 23.00%            | 81.90%                   |
>
> These findings clearly demonstrate the value of federated learning in fine-tuning, especially given the limitations of local data in terms of volume and diversity.
>
> Regarding the iterative training process, our analysis (Figure 7) reveals that a single iteration is insufficient for model convergence. The training curve for the Google Command dataset using GPT-FL shows that over 500 rounds are necessary to achieve reasonable performance, countering the assertion that one iteration would be adequate.

---

> ### Author Response · Authors · 2023-11-17
> **Response to Reviewer p1Ep (2/2)**
>
> ```
> Weakness 2: Second, unlike the claim in the paper, I do not believe that the proposed framework can deal with client drift effectively. If the distribution of synthetic data is significantly different from those of clients, then fine-tuning the global model might be the same as training a model from scratch locally by the clients.
> ```
>
> **Response**:
> The GPT-FL framework is still valid when generative models fail to generate accurate datasets. Our framework's reliance on generated data is not absolute; e.g., even with potential errors in the audio data, the combined use of synthetic and real data significantly enhances performance. In Section 4.2, we acknowledged the possibility of errors in synthetic audio data. However, GPT-FL's accuracy still notably improved compared to exclusive reliance on FL, showcasing the robustness of our method even in the presence of such errors. As emphasized in the benefits of GPT-FL and demonstrated in Table 4, consistent results highlighted the significant performance gains from FL fine-tuning. Regardless of synthetic data modality or quality, FL fine-tuning leads to significant performance gains, surpassing models trained solely with FL or FL combined with synthetic training by a large margin.
>
> Also, it is worth noting that some previous studies [1,2] have shown the powerfulness of synthetic data. In one notable study [1], synthetic ImageNet data were generated based on the dataset's label names, and a downstream model was centrally trained with this synthetic data and then tested on the real CIFAR-100 dataset. The remarkable performance observed in this study underscores that even when synthetic data diverges significantly from the test dataset's domain, it can still offer considerable value to the learning process. These findings align with our observations in the GPT-FL framework, further validating the potential benefits of incorporating synthetic data in federated learning contexts.
>
> We have also added two new experiments regarding more domain-specific tasks with Food 101[3] and the COVID-19 X-rays[4] dataset in the revised Appendix.
>
> |                 | 3x Synthetic | FedAvg | GPT-FL w/ FedAvg |
> |-----------------|--------------|--------|------------------|
> | COVID-19 X-rays | 37.71%       | 78.36% | 94.65%           |
> | Food 101        | 35.14%       | 43.25% | 70.57%           |
>
> The results demonstrate that even though the synthetic data has significant differences compared to actual data, GPT-FL still provides performance benefits compared to exclusive reliance on FL.
>
> [1] He, Ruifei, et al. "Is synthetic data from generative models ready for image recognition?." ICLR 2023.
>
> [2] Shipard, Jordan, et al. "Diversity is Definitely Needed: Improving Model-Agnostic Zero-shot Classification via Stable Diffusion." Proceedings of the IEEE/CVF Conference on Computer Vision and Pattern Recognition. 2023.
>
> [3] Bossard, Lukas et al. “Food-101 - Mining Discriminative Components with Random Forests.” European Conference on Computer Vision (2014).
>
> [4] Chowdhury, Muhammad Enamul Hoque et al. “Can AI Help in Screening Viral and COVID-19 Pneumonia?” IEEE Access 8 (2020): 132665-132676.
>
> ---
> If these responses have satisfactorily addressed your concerns, please consider increasing your score. Thank you!

---

> ### Author Response · Authors · 2023-11-19
>
> Dear Reviewer, we want to thank you again for your valuable comments. Feel free to let us know if you have any questions on our response.

---

> ### Comment · Reviewer_p1Ep · 2023-11-21
> **Thank you for the response**
>
> After reading the response I still believe that the motivation of the proposed approach is not strong. The novelty of this work is initializing the model for federated learning and the proposed approach seems to be decoupled from federated learning. Therefore, it is not clear that how it can help the quality of model learned at the end by cooperation of clients. Probably this work can be effective for the cases where the model initialization affect the convergence. This should be discussed in the paper and it would be great if the paper provides some analysis beyond experiments to convince readers that the proposed initialization can work well compared to other alternatives.

---

> ### Author Response · Authors · 2023-11-22
>
> Thank you for your reply. Here is our response.
>
> ---
> ```
> W1: After reading the response I still believe that the motivation of the proposed approach is not strong. The novelty of this work is initializing the model for federated learning and the proposed approach seems to be decoupled from federated learning.
> ```
>
> **Response**: Model initialization is not only an important part of federated learning but also a critical factor influencing its overall performance. Recent trends in federated learning research, as evidenced by studies published at top AI venues like ICLR [1,2,3] have increasingly focused on the significance of model initialization for federated learning. Our work aligns with these studies and proposes new techniques for model initialization in federated learning.
>
> ---
>
> ```
> W2: Therefore, it is not clear that how it can help the quality of model learned at the end by cooperation of clients.
> ```
>
> To demonstrate how our model initialization helps the quality of the model learned at the end by cooperation of clients, we compare the performance of the model initialized through GPT-FL followed by iterative learning by cooperation of clients vs. model randomly initialized followed by iterative learning by cooperation of clients. We compare their performance on five datasets: CIFAR-10, CIFAR-100, Flowers 102,  Google Command, and ESC-50.
>
> |                | Model randomly initialized followed by iterative learning by cooperation of clients | Model initialized through GPT-FL followed by iterative learning by cooperation of clients |   |   |
> |----------------|-------------------------------------------------------------------------------------|-------------------------------------------------------------------------------------------|---|---|
> | CIFAR-10       | 64.48%                                                                              | 81.38%                                                                                    |   |   |
> | CIFAR-100      | 25.89%                                                                              | 62.83%                                                                                    |   |   |
> | Flowers102     | 30.30%                                                                              | 70.56%                                                                                    |   |   |
> | Google Command | 73.68%                                                                              | 81.90%                                                                                    |   |   |
> | ESC-50         | 22.76%                                                                              | 41.80%                                                                                    |   |   |
>
> As shown, our initialization method consistently outperforms random initialization adopted in standard federated learning methods across all the five datasets, which demonstrates the superiority of our proposed model initialization approach.
>
> These results are actually already presented in Table 4 columns 2 and 4.
>
> ---
>
> ```
> W3: Probably this work can be effective for the cases where the model initialization affect the convergence. This should be discussed in the paper and it would be great if the paper provides some analysis beyond experiments to convince readers that the proposed initialization can work well compared to other alternatives.
> ```
>
> **Response**: Our results shown in Figure 6 and 7 demonstrate that our proposed model initialization approach is able to significantly reduce the convergence time. On page 9 of our paper, we analyze the gradient diversity for both random-initialized FedAvg and GPT-FL, which also aligns with the findings reported in [2] on the importance of model initialization in FL. Moreover, [3] already provided a theoretical analysis of why the model initialized with the pre-trained model can reduce the convergence time. We will add the reference to [3] in our revised version.
>
> [1] Tan, Yue et al. “Federated Learning from Pre-Trained Models: A Contrastive Learning Approach.” NeurIPS 2022
>
> [2] Nguyen, John et al. “Where to Begin? On the Impact of Pre-Training and Initialization in Federated Learning.” ICLR 2023
>
> [3] Chen, Hong-You et al. “On the Importance and Applicability of Pre-Training for Federated Learning.” ICLR 2023
>
> ---
>
> We hope your concerns could be resolved and the rating of the paper can be increased accordingly. Thank you!

---

### Official Review · Reviewer_5fiA · 2023-11-03

**Soundness:** 2 fair
**Presentation:** 3 good
**Contribution:** 2 fair
**Rating:** 5
**Confidence:** 3

**Summary:**

This paper proposes GPT-FL, which leverages generative pretrained models to generate synthetic data for training a downstream model for FL. The experimental results show the remarkable performance of GPT-FL when compared with state-of-the-art FL methods.

**Strengths:**

The writing is mostly clear. The proposed method significantly outperforms state-of-the-art FL methods.

**Weaknesses:**

My main concern is that GPT-FL may significantly increase the burden of the central server. Notice that in the standard FL, the server only needs to aggregate parameters received from clients. However, GPT-FL needs to run pre-trained models on the central server, which requires additional computational cost.

**Questions:**

1. Which pretained models are used in the experiments?
2. I think the experiments may compare the time and memory overhead used by GPT-FL and standard FL methods.

---

> ### Author Response · Authors · 2023-11-17
> **Response to Reviewer 5fiA (1/1)**
>
> Thank you for your time and suggestions. Here are our responses.
>
> ---
> ```
> Weakness 1: My main concern is that GPT-FL may significantly increase the burden of the central server. Notice that in the standard FL, the server only needs to aggregate parameters received from clients. However, GPT-FL needs to run pre-trained models on the central server, which requires additional computational cost.
> ```
> **Response**: Central servers in FL are cloud servers which have significantly more computational power compared to clients. The additional computational cost incurred by running pre-trained models is minimal compared to how many computational resources a central server has. In standard FL, the server only needs to aggregate parameters. Although this incurs little computational cost, the communication cost of collecting parameters from clients through hundreds of iterations is the bottleneck. In contrast, GPT-FL significantly cuts the communication cost. This is the significant benefit of running the pre-trained models on the central server.
>
> Utilizing this server capability is crucial for mitigating client drift, a limitation of traditional FL methods as evidenced by our results in Table 2 and supported by prior research (referenced in Table 2). The additional computational cost is a justified trade-off for the enhanced performance and drift mitigation provided by GPT-FL. Our approach is aligned with current trends in leveraging server power for more effective FL, such as the baselines we listed in Table 1 and Table 2.
>
> ---
> ```
> Question 1: Which pretained models are used in the experiments?
> ```
> **Response**: As we stated in Section 3.2 of the paper, we utilize the state-of-the-art Latent Diffusion Model loaded with Stable Diffusion V2.1 weights to generate synthetic images for image-based FL applications, and we utilize the state-of-the-art SpeechT5 model for text-to-speech and AudioLDM model for text-to-audio to generate synthetic speech and audio data, respectively.
>
> ---
> ```
> Question 2: I think the experiments may compare the time and memory overhead used by GPT-FL and standard FL methods.
> ```
> **Response**: It is worth noting that GPT-FL does not alter the standard FL framework, which means that each client in GPT-FL would not have an additional computational or communicational burden, even any additional hyper-parameters beyond the standard FL framework.
>
> As demonstrated in Figures 2, 3, and 4, GPT-FL requires significantly fewer communication rounds compared to standard FL methods for achieving the same target accuracy. This reduction in communication rounds translates to considerable computational savings for the clients. The primary reason behind this efficiency is the offloading of a substantial portion of computation to the server side, specifically through the pre-training of the downstream model. By leveraging the computational power of the server in this manner, GPT-FL not only maintains client-side efficiency but also enhances the overall performance and speed of the federated learning process.
>
> ---
>
> If these responses have satisfactorily addressed your concerns, please consider increasing your score. Thank you!

---

> ### Author Response · Authors · 2023-11-19
>
> Dear Reviewer, we want to thank you again for your valuable comments. Feel free to let us know if you have any questions on our response.

---

### Meta-Review · Area_Chair_Rvc3 · 2023-12-17

**Metareview:**

In federated learning, final model quality can be improved if the "starting model at time step 0 from the server" is already initialized to be a good model. This paper's method takes in labels from clients, makes them into prompts to feed into GPT, uses the resulting data to train the initial model, and then uses this model as the initialization for FL. This is shown to be better than (a) methods that just use existing datasets for initialization, and (b) methods that use less capable generative models for making the synthetic data.


Strength: clear writing.

Weakness: low novelty; it is known that GPT can be effectively used to make synthetic data given a prescription; this paper just translates the gains of that into federated learning.

**Justification For Why Not Higher Score:**

The algorithmic novelty of this paper is minimal; see the meta review above for context.

**Justification For Why Not Lower Score:**

it is the lowest score.

---

### Decision · Program_Chairs · 2024-01-16

Reject